# On the Convergence of Adaptive Gradient Methods for Nonconvex Optimization

**Dongruo Zhou**[*]                                                                 *dz13@iu.edu*
*Indiana University*

**Jinghui Chen**[*]                                                                 *jzc5917@psu.edu*
*The Pennsylvania State University*

**Yuan Cao**[*]                                                                     *yuancao@hku.hk*
*The University of Hong Kong*

**Ziyan Yang**                                                                      *zy47@rice.edu*
*Rice University*

**Quanquan Gu**                                                                     *qgu@cs.ucla.edu*
*University of California, Los Angeles*

**Reviewed on OpenReview:** *https://openreview.net/forum?id=Gh0cxhbz3c*

## Abstract

Adaptive gradient methods are workhorses in deep learning. However, the convergence guarantees of adaptive gradient methods for nonconvex optimization have not been thoroughly studied. In this paper, we provide a fine-grained convergence analysis for a general class of adaptive gradient methods including AMSGrad, RMSProp and AdaGrad. For smooth nonconvex functions, we prove that adaptive gradient methods in expectation converge to a first-order stationary point. Our convergence rate is better than existing results for adaptive gradient methods in terms of dimension. In addition, we also prove high probability bounds on the convergence rates of AMSGrad, RMSProp as well as AdaGrad, which have not been established before. Our analyses shed light on better understanding the mechanism behind adaptive gradient methods in optimizing nonconvex objectives.

## 1 Introduction

Stochastic gradient descent (SGD) (Robbins & Monro, 1951) and its variants have been widely used in training deep neural networks. Among those variants, adaptive gradient methods (AdaGrad) (Duchi et al., 2011; McMahan & Streeter, 2010), which scale each coordinate of the gradient by a function of past gradients, can achieve better performance than vanilla SGD in practice when the gradients are sparse. An intuitive explanation for the success of AdaGrad is that it automatically adjusts the learning rate for each feature based on the partial gradient, which accelerates the convergence. However, AdaGrad was later found to demonstrate degraded performance especially in cases where the loss function is nonconvex or the gradient is dense, due to rapid decay of learning rate. This problem is especially exacerbated in deep learning due to the huge number of optimization variables. To overcome this issue, RMSProp (Tieleman & Hinton, 2012) was proposed to use exponential moving average rather than the arithmetic average to scale the gradient, which mitigates the rapid decay of the learning rate. Kingma & Ba (2014) proposed an adaptive momentum estimation method (Adam), which incorporates the idea of momentum (Polyak, 1964; Sutskever et al., 2013) into RMSProp. Other related algorithms include AdaDelta (Zeiler, 2012) and Nadam (Dozat, 2016), which combine the idea of the exponential moving average of the historical gradients, Polyak's heavy

---

[*]Equal Contribution

ball (Polyak, 1964) and Nesterov's accelerated gradient descent (Nesterov, 2013). Recently, by revisiting the original convergence analysis of Adam, Reddi et al. (2018) found that for some handcrafted simple convex optimization problem, Adam does not even converge to the global minimizer. In order to address this convergence issue of Adam, Reddi et al. (2018) proposed a new variant of the Adam algorithm named AMSGrad, which has guaranteed convergence in the convex setting. The update rule of AMSGrad is as follows[1]:

$$\mathbf{x}_{t+1} = \mathbf{x}_t - \alpha_t \frac{\mathbf{m}_t}{\sqrt{\widehat{\mathbf{v}}_t} + \epsilon}, \quad \widehat{\mathbf{v}}_t = \max(\widehat{\mathbf{v}}_{t-1}, \mathbf{v}_t), \tag{1.1}$$

where $\alpha_t > 0$ is the step size, $\epsilon$ is a small number to ensure numerical stability, $\mathbf{x}_t \in \mathbb{R}^d$ is the iterate in the $t$-th iteration, and $\mathbf{m}_t, \mathbf{v}_t \in \mathbb{R}^d$ are the exponential moving averages of the gradient and the squared gradient at the $t$-th iteration respectively: [2]

$$\mathbf{m}_t = \beta_1 \mathbf{m}_{t-1} + (1 - \beta_1)\mathbf{g}_t, \quad \mathbf{v}_t = \beta_2 \mathbf{v}_{t-1} + (1 - \beta_2)\mathbf{g}_t^2. \tag{1.2}$$

Here $\beta_1, \beta_2 \in [0, 1]$ are algorithm hyperparameters, and $\mathbf{g}_t$ is the stochastic gradient at $\mathbf{x}_t$.

Despite the successes of adaptive gradient methods for training deep neural networks, the convergence guarantees for these algorithms are mostly restricted to online convex optimization (Duchi et al., 2011; Kingma & Ba, 2014; Reddi et al., 2018). Therefore, there is a huge gap between existing online convex optimization guarantees for adaptive gradient methods and the empirical successes of adaptive gradient methods in nonconvex optimization. In order to bridge this gap, there are a few recent attempts to prove the nonconvex optimization guarantees for adaptive gradient methods. More specifically, Basu et al. (2018) proved the convergence rate of RMSProp and Adam when using deterministic gradient rather than stochastic gradient. Li & Orabona (2018) proved the convergence rate of AdaGrad, assuming the gradient is $L$-Lipschitz continuous. Ward et al. (2018) proved the convergence rate of AdaGrad-Norm where the moving average of the norms of the gradient vectors is used to adjust the gradient vector in both deterministic and stochastic settings for smooth nonconvex functions. Nevertheless, the convergence guarantees in Basu et al. (2018); Ward et al. (2018) are still limited to simplified algorithms. Another attempt to obtain the convergence rate under stochastic setting is prompted recently by Zou & Shen (2018), in which they only focus on the condition when the momentum vanishes. Chen et al. (2018a) studies the convergence properties of adaptive gradient methods in the nonconvex setting, however, its convergence rate has a quadratic dependency on the problem dimension $d$. Défossez et al. (2020) proves the convergence of Adam and Adagrad in nonconvex smooth optimization under the assumption of almost sure uniform bound on the $L_\infty$ norm of the gradients. In this paper, we provide a fine-grained convergence analysis of the adaptive gradient methods. In particular, we analyze several representative adaptive gradient methods, i.e., AMSGrad (Reddi et al., 2018), which fixed the non-convergence issue in Adam and the RMSProp (fixed version via (Reddi et al., 2018)), and prove its convergence rate for smooth nonconvex objective functions in the stochastic optimization setting. Moreover, existing theoretical guarantees for adaptive gradient methods are mostly bounds in expectation over the randomness of stochastic gradients, and are therefore only on-average convergence guarantees. In practice, however, the optimization algorithm is usually only run once, and therefore the performance cannot be guaranteed by the in-expectation bounds. To deal with this problem, we also provide high probability convergence rates for AMSGrad and RMSProp, which can characterize the performance of the algorithms on a single run.

## 1.1 Our Contributions

The main contributions of our work are as follows:

---

[1]With slight abuse of notation, here we denote by $\sqrt{\mathbf{v}_t}$ the element-wise square root of the vector $\mathbf{v}_t$, $\mathbf{m}_t/\sqrt{\mathbf{v}_t}$ the element-wise division between $\mathbf{m}_t$ and $\sqrt{\mathbf{v}_t}$, and $\max(\widehat{\mathbf{v}}_{t-1}, \mathbf{v}_t)$ the element-wise maximum between $\widehat{\mathbf{v}}_{t-1}$ and $\mathbf{v}_t$.
[2]We denote by $\mathbf{g}_t^2$ the element-wise square of the vector $\mathbf{g}_t$.

- We prove that the convergence rate of AMSGrad to a stationary point for stochastic nonconvex optimization is

$$O\left(\frac{d^{1/2}}{T^{3/4-s/2}} + \frac{d}{T}\right),\tag{1.3}$$

  when $\|\mathbf{g}_{1:T,i}\|_2 \leq G_\infty T^s$. Here $\mathbf{g}_{1:T,i} = [g_{1,i}, g_{2,i}, \ldots, g_{T,i}]^\top$ with $\{\mathbf{g}_t\}_{t=1}^T$ being the stochastic gradients satisfying $\|\mathbf{g}_t\|_\infty \leq G_\infty$, and $s \in [0, 1/2]$ is a parameter that characterizes the growth rate of the cumulative stochastic gradient $\mathbf{g}_{1:T,i}$.

- Our result implies that the worst case (i.e., $s = 1/2$) convergence rate for AMSGrad is

$$O\left(\sqrt{\frac{d}{T}} + \frac{d}{T}\right),$$

  which has a better dependence on the dimension $d$ and $T$ than the convergence rate proved in Chen et al. (2018a), i.e.,

$$O\left(\frac{\log T + d^2}{\sqrt{T}}\right).$$

- We also establish high probability bounds for adaptive gradient methods. To the best of our knowledge, it is the first high probability convergence guarantees for AMSGrad and RMSProp for nonconvex stochastic optimization.

**Notations:** scalars are denoted by lower case letters, vectors by lower case bold face letters, and matrices by upper case bold face letters. For a vector $\mathbf{x} = [x_i] \in \mathbb{R}^d$, we denote the $\ell_p$ norm ($p \geq 1$) of $\mathbf{x}$ by $\|\mathbf{x}\|_p = \left(\sum_{i=1}^d |x_i|^p\right)^{1/p}$, the $\ell_\infty$ norm of $\mathbf{x}$ by $\|\mathbf{x}\|_\infty = \max_{i=1}^d |x_i|$. For a sequence of vectors $\{\mathbf{g}_j\}_{j=1}^t$, we denote by $g_{j,i}$ the $i$-th element in $\mathbf{g}_j$. We also denote $\mathbf{g}_{1:t,i} = [g_{1,i}, g_{2,i}, \ldots, g_{t,i}]^\top$. With slightly abuse of notation, for any two vectors $\mathbf{a}$ and $\mathbf{b}$, we denote $\mathbf{a}^2$ as the element-wise square, $\mathbf{a}^p$ as the element-wise power operation, $\mathbf{a}/\mathbf{b}$ as the element-wise division and $\max(\mathbf{a}, \mathbf{b})$ as the element-wise maximum. For a matrix $\mathbf{A} = [A_{ij}] \in \mathbb{R}^{d\times d}$, we define $\|\mathbf{A}\|_{1,1} = \sum_{i,j=1}^d |A_{ij}|$ and $\|\mathbf{A}\|_{\infty,\infty} = \max_{i,j=1}^d |A_{ij}|$. Given two sequences $\{a_n\}$ and $\{b_n\}$, we write $a_n = O(b_n)$ if there exists a constant $0 < C < +\infty$ such that $a_n \leq C b_n$. We use notation $\widetilde{O}(\cdot)$ to hide logarithmic factors.

## 2 Related Work

Here we review other related work that is not covered before.

**Adaptive gradient methods**: Mukkamala & Hein (2017) proposed SC-Adagrad and SC-RMSprop, which derives logarithmic regret bounds for strongly convex functions. Chen et al. (2018b) proposed SADAGRAD for solving stochastic strongly convex optimization and more generally stochastic convex optimization that satisfies the second order growth condition. Zaheer et al. (2018) studied the effect of adaptive denominator constant $\epsilon$ and minibatch size in the convergence of adaptive gradient methods. Zou et al. (2019) presented an easy-to-check sufficient condition to guarantee the convergences of Adam and AMSGrad in the nonconvex stochastic setting. Chen et al. (2020) proposed a partially adaptive gradient method and proved its convergence in nonconvex settings. Alacaoglu et al. (2020) proposed a new framework to derive data-dependent regret bounds with a constant momentum parameter in various settings.

**Nonconvex Stochastic Optimization**: Ghadimi & Lan (2013) proposed a randomized stochastic gradient (RSG) method, and proved its $O(1/\sqrt{T})$ convergence rate to a stationary point. Ghadimi & Lan (2016) proposed an randomized stochastic accelerated gradient (RSAG) method, which achieves $O(1/T + \sigma^2/\sqrt{T})$ convergence rate, where $\sigma^2$ is an upper bound on the variance of the stochastic gradient. Motivated by

---

[3]To be precise, Li & Orabona (2020) studies a delayed AdaGrad algorithm with momentum.

Table 1: Comparison of convergence rate of AMSGrad and AdaGrad in terms of the convergence types and assumptions by different works in the nonconvex smooth setting. Here $T$ denotes the total number of iterations and $d$ is the dimension.

| | Conv. Rate | Conv. Type | Assumptions |
|---|---|---|---|
| **AMSGrad** | | | |
| Chen et al. (2018a) | $O\left(\frac{\log T + d^2}{\sqrt{T}}\right)$ | in-expectation | smoothness, bounded gradient |
| Alacaoglu et al. (2020) | $O\left(\frac{d \log T}{\sqrt{T}}\right)$ | in-expectation | smoothness, bounded gradient |
| Ours (worst case, i.e., $s = 1/2$) | $O\left(\sqrt{\frac{d}{T}} + \frac{d}{T}\right)$ | in-expectation | smoothness, bounded gradient |
| Ours (worst case, i.e., $s = 1/2$) | $O\left(\sqrt{\frac{d}{T}} + \frac{d}{T}\right)$ | high probability | smoothness, bounded gradient, $\nabla f(\mathbf{x}, \xi) - \nabla f(\mathbf{x})$ is a sub-Gaussian vector |
| **AdaGrad** | | | |
| Défossez et al. (2020) | $O\left(\frac{1}{\sqrt{T}} + \frac{d}{\sqrt{T}}\right)$ | in-expectation | smoothness, bounded gradient |
| Li & Orabona (2020)[3] | $O\left(\frac{d}{\sqrt{T}}\right)$ | high probability | smoothness, $\|\nabla f(\mathbf{x}, \xi) - \nabla f(\mathbf{x})\|_2$ is sub-Gaussian |
| Ours (worst case, i.e., $s = 1/2$) | $O\left(\sqrt{\frac{d}{T}} + \frac{d}{T}\right)$ | in-expectation | smoothness, bounded gradient |
| Ours (worst case, i.e., $s = 1/2$) | $O\left(\sqrt{\frac{d}{T}} + \frac{d}{T}\right)$ | high probability | smoothness, bounded gradient, $\nabla f(\mathbf{x}, \xi) - \nabla f(\mathbf{x})$ is a sub-Gaussian vector |

the success of stochastic momentum methods in deep learning (Sutskever et al., 2013), Yang et al. (2016) provided a unified convergence analysis for both stochastic heavy-ball method and the stochastic variant of Nesterov's accelerated gradient method, and proved $O(1/\sqrt{T})$ convergence rate to a stationary point for smooth nonconvex functions. Reddi et al. (2016); Allen-Zhu & Hazan (2016) proposed variants of stochastic variance-reduced gradient (SVRG) method (Johnson & Zhang, 2013) that is provably faster than gradient descent in the nonconvex finite-sum setting. Lei et al. (2017) proposed a stochastically controlled stochastic gradient (SCSG), which further improves convergence rate of SVRG for finite-sum smooth nonconvex optimization. Recently, Zhou et al. (2018) proposed a new algorithm called stochastic nested variance-reduced gradient (SNVRG), which achieves strictly better gradient complexity than both SVRG and SCSG for finite-sum and stochastic smooth nonconvex optimization.

**High Probability Bounds**: There are only a few works on the high probability convergence results. Kakade & Tewari (2009) proved high probability bounds for the PEGASOS algorithm via Freeman's inequality. Harvey et al. (2019a;b) proved convergence bounds for non-smooth, strongly convex case via generalized Freeman's inequality. Jain et al. (2019) makes the last iterate of SGD information theoretically optimal by providing a high probability bound. Li & Orabona (2020) presented a high probability analysis for Delayed AdaGrad algorithm with momentum in the smooth nonconvex setting.

For the ease of comparison, we summarize the convergence rates of adaptive gradient methods derived in different works in Table 1, along with the convergence types and corresponding assumptions.

## 3 Algorithms

We mainly consider the following three algorithms: AMSGrad (Reddi et al., 2018), a corrected version of RMSProp (Tieleman & Hinton, 2012; Reddi et al., 2018), and AdaGrad (Duchi et al., 2011).

---

**Algorithm 1** AMSGrad (Reddi et al., 2018)

---

**Require:** Initial point $\mathbf{x}_1$, step size $\{\alpha_t\}_{t=1}^T$, adaptive gradient parameters $\beta_1$, $\beta_2$, $\epsilon$.
1: $\mathbf{m}_0 \leftarrow 0$, $\widehat{\mathbf{v}}_0 \leftarrow 0$, $\mathbf{v}_0 \leftarrow 0$
2: **for** $t = 1$ to $T$ **do**
3: $\quad \mathbf{g}_t = \nabla f(\mathbf{x}_t, \xi_t)$
4: $\quad \mathbf{m}_t = \beta_1 \mathbf{m}_{t-1} + (1 - \beta_1)\mathbf{g}_t$
5: $\quad \mathbf{v}_t = \beta_2 \mathbf{v}_{t-1} + (1 - \beta_2)\mathbf{g}_t^2$
6: $\quad \widehat{\mathbf{v}}_t = \max(\widehat{\mathbf{v}}_{t-1}, \mathbf{v}_t)$
7: $\quad \mathbf{x}_{t+1} = \mathbf{x}_t - \alpha_t \widehat{\mathbf{V}}_t^{-1/2}\mathbf{m}_t$ with $\widehat{\mathbf{V}}_t = \mathrm{diag}(\widehat{\mathbf{v}}_t + \epsilon)$
8: **end for**
**Ensure:** Choose $\mathbf{x}_{\mathrm{out}}$ from $\{\mathbf{x}_t\}$, $2 \leq t \leq T$ with probability $\alpha_{t-1}/\sum_{i=1}^{T-1} \alpha_i$.

---

The AMSGrad algorithm is originally proposed by Reddi et al. (2018) to fix the non-convergence issue in the original Adam optimizer (Kingma & Ba, 2014). Specifically, in Algorithm 1, the effective learning rate of AMSGrad is $\alpha_t \widehat{\mathbf{V}}_t^{-1/2}$ where $\widehat{\mathbf{V}}_t = \mathrm{diag}(\widehat{\mathbf{v}}_t)$, while in original Adam, the effective learning rate is $\alpha_t \mathbf{V}_t^{-1/2}$ where $\mathbf{V}_t = \mathrm{diag}(\mathbf{v}_t)$. This choice of effective learning rate guarantees that it is non-increasing and thus fix the possible convergence issue. In Algorithm 2, we present a variant of RMSProp (Tieleman & Hinton, 2012) (adding the max step according to Reddi et al. (2018)) where the effective learning rate is also set as $\alpha_t \widehat{\mathbf{V}}_t^{-1/2}$.

---

**Algorithm 2** RMSProp (Tieleman & Hinton, 2012) (modified according to Reddi et al. (2018))

---

**Require:** Initial point $\mathbf{x}_1$, step size $\{\alpha_t\}_{t=1}^T$, adaptive gradient parameters $\beta, \epsilon$.
1: $\widehat{\mathbf{v}}_0 \leftarrow 0$, $\mathbf{v}_0 \leftarrow 0$
2: **for** $t = 1$ to $T$ **do**
3: $\quad \mathbf{g}_t = \nabla f(\mathbf{x}_t, \xi_t)$
4: $\quad \mathbf{v}_t = \beta \mathbf{v}_{t-1} + (1 - \beta)\mathbf{g}_t^2$
5: $\quad \widehat{\mathbf{v}}_t = \max(\widehat{\mathbf{v}}_{t-1}, \mathbf{v}_t)$
6: $\quad \mathbf{x}_{t+1} = \mathbf{x}_t - \alpha_t \widehat{\mathbf{V}}_t^{-1/2}\mathbf{g}_t$ with $\widehat{\mathbf{V}}_t = \mathrm{diag}(\widehat{\mathbf{v}}_t + \epsilon)$
7: **end for**
**Ensure:** Choose $\mathbf{x}_{\mathrm{out}}$ from $\{\mathbf{x}_t\}$, $2 \leq t \leq T$ with probability $\alpha_{t-1}/\sum_{i=1}^{T-1} \alpha_i$.

---

**Algorithm 3** AdaGrad (Duchi et al., 2011)

---

**Require:** Initial point $\mathbf{x}_1$, step size $\{\alpha_t\}_{t=1}^T$, adaptive gradient parameter $\epsilon$.
1: $\widehat{\mathbf{v}}_0 \leftarrow 0$
2: **for** $t = 1$ to $T$ **do**
3: $\quad \mathbf{g}_t = \nabla f(\mathbf{x}_t, \xi_t)$
4: $\quad \widehat{\mathbf{v}}_t = \widehat{\mathbf{v}}_{t-1} + \mathbf{g}_t^2$
5: $\quad \mathbf{x}_{t+1} = \mathbf{x}_t - \alpha_t \widehat{\mathbf{V}}_t^{-1/2}\mathbf{g}_t$ with $\widehat{\mathbf{V}}_t = \mathrm{diag}(\widehat{\mathbf{v}}_t + \epsilon)$
6: **end for**
**Ensure:** Choose $\mathbf{x}_{\mathrm{out}}$ from $\{\mathbf{x}_t\}$, $2 \leq t \leq T$ with probability $\alpha_{t-1}/\sum_{i=1}^{T-1} \alpha_i$.

---

In Algorithm 3 we further present the AdaGrad algorithm (Duchi et al., 2011), which adopts the summation of past stochastic gradient squares instead of the running average to compute the effective learning rate.

# 4 Convergence Results in Expectation

In this section, we present our main results on the convergence of AMSGrad, RMSProp and AdaGrad. We study the following stochastic nonconvex optimization problem

$$\min_{\mathbf{x} \in \mathbb{R}^d} f(\mathbf{x}) := \mathbb{E}_\xi \big[ f(\mathbf{x}; \xi) \big],$$

where $\xi$ is a random variable satisfying certain distribution, $f(\mathbf{x}; \xi) : \mathbb{R}^d \to \mathbb{R}$ is a $L$-smooth nonconvex function. In the stochastic setting, one cannot directly access the full gradient of $f(\mathbf{x})$. Instead, one can only get unbiased estimators of the gradient of $f(\mathbf{x})$, which is $\nabla f(\mathbf{x}; \xi)$. This setting has been studied in Ghadimi & Lan (2013; 2016).

**Assumption 4.1 (Bounded Gradient)** $f(\mathbf{x}) = \mathbb{E}_\xi f(\mathbf{x}; \xi)$ has $G_\infty$-bounded stochastic gradient. That is, for any $\xi$, we assume that $\|\nabla f(\mathbf{x}; \xi)\|_\infty \leq G_\infty$.

It is worth mentioning that Assumption 4.1 is slightly weaker than the $\ell_2$-boundedness assumption $\|\nabla f(\mathbf{x}; \xi)\|_2 \leq G_2$ used in Reddi et al. (2016); Chen et al. (2018a). Since $\|\nabla f(\mathbf{x}; \xi)\|_\infty \leq \|\nabla f(\mathbf{x}; \xi)\|_2 \leq \sqrt{d}\|\nabla f(\mathbf{x}; \xi)\|_\infty$, the $\ell_2$-boundedness assumption implies Assumption 4.1 with $G_\infty = G_2$. Meanwhile, $G_\infty$ will be tighter than $G_2$ by a factor of $\sqrt{d}$ when each coordinate of $\nabla f(\mathbf{x}; \xi)$ almost equals to each other.

**Assumption 4.2 ($L$-smooth)** $f(\mathbf{x}) = \mathbb{E}_\xi f(\mathbf{x}; \xi)$ is $L$-smooth: for any $\mathbf{x}, \mathbf{y} \in \mathbb{R}^d$, we have

$$\left| f(\mathbf{x}) - f(\mathbf{y}) - \langle \nabla f(\mathbf{y}), \mathbf{x} - \mathbf{y} \rangle \right| \leq \frac{L}{2}\|\mathbf{x} - \mathbf{y}\|_2^2.$$

Assumption 4.2 is a standard assumption in the analysis of gradient-based algorithms. It is equivalent to the $L$-gradient Lipschitz condition, which is often written as $\|\nabla f(\mathbf{x}) - \nabla f(\mathbf{y})\|_2 \leq L\|\mathbf{x} - \mathbf{y}\|_2$.

We are now ready to present our main result.

**Theorem 4.3 (AMSGrad)** Suppose $\beta_1 < \beta_2^{1/2}$, $\alpha_t = \alpha$ and $\|\mathbf{g}_{1:T,i}\|_2 \leq G_\infty T^s$ for $t = 1, \ldots, T, 0 \leq s \leq 1/2$. Then under Assumptions 4.1 and 4.2, the iterates $\mathbf{x}_t$ of AMSGrad satisfy that

$$\frac{1}{T-1}\sum_{t=2}^{T} \mathbb{E}\left[\|\nabla f(\mathbf{x}_t)\|_2^2\right] \leq \frac{M_1}{T\alpha} + \frac{M_2 d}{T} + \frac{\alpha M_3 d}{T^{1/2-s}}, \tag{4.1}$$

where $\{M_i\}_{i=1}^3$ are defined as follows:

$$M_1 = 2(G_\infty + \sqrt{\epsilon})\Delta, \quad M_2 = \frac{2G_\infty^2(G_\infty + \sqrt{\epsilon})\epsilon^{-1/2}}{1 - \beta_1} + 2G_\infty(G_\infty + \sqrt{\epsilon}),$$

$$M_3 = \frac{2LG_\infty(G_\infty + \sqrt{\epsilon})}{\epsilon^{1/2}(1 - \beta_2)^{1/2}(1 - \beta_1/\beta_2^{1/2})}\left(1 + \frac{2\beta_1^2}{1 - \beta_1}\right),$$

and $\Delta = f(\mathbf{x}_1) - \inf_{\mathbf{x}} f(\mathbf{x})$.

Note that in Theorem 4.3 we have a condition that $\|\mathbf{g}_{1:T,i}\|_2 \leq G_\infty T^s$. Here $s$ characterizes the growth rate of $\mathbf{g}_{1:T,i}$, i.e., the cumulative stochastic gradient (Liu et al., 2019). In the worse case where the stochastic gradients are not sparse, we have $s = 1/2$, while in practice when the stochastic gradients are sparse, we have $s < 1/2$.

**Remark 4.4** If we choose $\alpha = \Theta\left(d^{1/2}T^{1/4+s/2}\right)^{-1}$, then (4.1) implies that AMSGrad achieves

$$O\left(\frac{d^{1/2}}{T^{3/4-s/2}} + \frac{d}{T}\right)$$

convergence rate. In the worst case when $s = 1/2$, this result matches the convergence rate of nonconvex SGD (Ghadimi & Lan, 2016). For the dimension dependence, it is not directly comparable since they made a different stochastic noise assumption (they assumed the stochastic gradient is $\sigma$-subGaussian w.r.t. the $\ell_2$ norm of the gradient). By directly translating their assumption to ours (to replace $\sigma$ with $\sqrt{d}G_\infty$), we can obtain a $\sqrt{d/T}$ dominant term in their convergence result, which matches our convergence rate. Note that Chen et al. (2018a) also provided a similar bound for AMSGrad that

$$\frac{1}{T-1}\sum_{t=2}^{T} \mathbb{E}\left[\|\nabla f(\mathbf{x}_t)\|_2^2\right] = O\left(\frac{\log T + d^2}{\sqrt{T}}\right).$$

*It can be seen that the dependence of d in their bound is quadratic, which is worse than the linear dependence suggested by (4.1). A recent work (Défossez et al., 2020) discussed the convergence issue of Adam by showing that the bound consists of a constant term and does not converge to zero. In comparison, our result for AMSGrad does not have such a constant term and converges to zero in a rate $O(d^{1/2}/T^{3/4-s/2})$. This suggests that the convergence issue of Adam is indeed fixed in AMSGrad.*

**Corollary 4.5 (A variant of RMSProp)** *Under the same conditions of Theorem 4.3, if $\alpha_t = \alpha$ and $\|\mathbf{g}_{1:T,i}\|_2 \le G_\infty T^s$ for $t = 1, \ldots, T, 0 \le s \le 1/2$, then the iterates $\mathbf{x}_t$ of RMSProp satisfy that*

$$\frac{1}{T-1} \sum_{t=2}^{T} \mathbb{E}\big[\|\nabla f(\mathbf{x}_t)\|_2^2\big] \le \frac{M_1}{T\alpha} + \frac{M_2 d}{T} + \frac{\alpha M_3 d}{T^{1/2-s}},$$

*where $\{M_i\}_{i=1}^3$ are defined as follows:*

$$M_1 = 2(G_\infty + \sqrt{\epsilon})\Delta, \ M_2 = 2G_\infty^2(G_\infty + \sqrt{\epsilon})\epsilon^{-1/2} + 2G_\infty(G_\infty + \sqrt{\epsilon}), \ M_3 = \frac{6LG_\infty(G_\infty + \sqrt{\epsilon})}{\epsilon^{1/2}(1-\beta)^{1/2}},$$

*and $\Delta = f(\mathbf{x}_1) - \inf_{\mathbf{x}} f(\mathbf{x})$.*

**Corollary 4.6 (AdaGrad)** *Under the same conditions of Theorem 4.3, if $\alpha_t = \alpha$ and $\|\mathbf{g}_{1:T,i}\|_2 \le G_\infty T^s$ for $t = 1, \ldots, T, 0 \le s \le 1/2$, then the the iterates $\mathbf{x}_t$ of AdaGrad satisfy that*

$$\frac{1}{T-1} \sum_{t=2}^{T} \mathbb{E}\big[\|\nabla f(\mathbf{x}_t)\|_2^2\big] \le \frac{M_1}{T\alpha} + \frac{M_2 d}{T} + \frac{\alpha M_3 d}{T^{1/2-s}},$$

*where $\{M_i\}_{i=1}^3$ are defined as follows:*

$$M_1 = 2(G_\infty + \sqrt{\epsilon})\Delta, \ M_2 = 2G_\infty^2(G_\infty + \sqrt{\epsilon})\epsilon^{-1/2} + 2G_\infty(G_\infty + \sqrt{\epsilon}), \ M_3 = 6LG_\infty(G_\infty + \sqrt{\epsilon})\epsilon^{-1/2},$$

*and $\Delta = f(\mathbf{x}_1) - \inf_{\mathbf{x}} f(\mathbf{x})$.*

Corollaries 4.5 and 4.6 imply that RMSProp and AdaGrad achieve the same rate of convergence as AMSGrad. In worst case where $s = 1/2$, both algorithms achieve $O(\sqrt{d/T} + d/T)$ convergence rate, which matches the convergences rate of nonconvex SGD given by Ghadimi & Lan (2016).

**Remark 4.7** *Défossez et al. (2020) gave a bound $O(\alpha^{-1}T^{-1/2} + (1 + \alpha)dT^{-1/2})$ for AdaGrad, which gives the following rate*

$$O\left(\frac{1}{\sqrt{T}} + \frac{d}{\sqrt{T}}\right)$$

*when $\alpha = 1$. Our result gives a faster rate in terms of the dependency in dimension d.*

## 5 Convergence Results with High Probability

In the previous section, we provide convergence results of the three adaptive gradient methods in expectation. These bounds can only guarantee the average performance of a large number of trials of the algorithm, but cannot rule out extremely bad solutions. What's more, for practical applications such as training deep neural networks, we often perform a single run of the algorithm since the training time can be fairly large. Hence, it is helpful to get high probability bounds which guarantee the performance of the algorithm on a single run. To overcome this limitation, in this section, we further establish high probability bounds on the convergence rate for AMSGrad, RMSProp and AdaGrad. We make the following additional assumption.

**Assumption 5.1** *The stochastic gradients are sub-Gaussian random vectors (Jin et al., 2019):*

$$\mathbb{E}_\xi[\exp(\langle \mathbf{v}, \nabla f(\mathbf{x}, \xi) - \nabla f(\mathbf{x})\rangle)] \le \exp(\|\mathbf{v}\|_2^2 \sigma^2/2)$$

*for all $\mathbf{v} \in \mathbb{R}^d$ and all $\mathbf{x}$.*

Assumption 5.1 is commonly considered when studying high probability bounds (Li & Orabona, 2020). It is weaker than Assumption B2 in Li & Orabona (2020): for the case when $\nabla f(\mathbf{x}, \xi) - \nabla f(\mathbf{x})$ is a standard Gaussian vector, $\sigma^2$ defined in Li & Orabona (2020) is of order $O(d)$, while $\sigma^2 = O(1)$ in our definition.

**Theorem 5.2 (AMSGrad)** *Suppose $\beta_1 < \beta_2^{1/2}$, $\alpha_t = \alpha \le \sigma^{-2}\epsilon/2$ and $\|\mathbf{g}_{1:T,i}\|_2 \le G_\infty T^s$ for $t = 1, \ldots, T, 0 \le s \le 1/2$. Then for any $\delta > 0$, under Assumptions 4.1, 4.2 and 5.1, with probability at least $1 - \delta$, the iterates $\mathbf{x}_t$ of AMSGrad satisfy that*

$$\frac{1}{T-1}\sum_{t=2}^{T}\|\nabla f(\mathbf{x}_t)\|_2^2 \le \frac{M_1}{T\alpha} + \frac{M_2 d}{T} + \frac{\alpha M_3 d}{T^{1/2-s}}, \tag{5.1}$$

*where $\{M_i\}_{i=1}^3$ are defined as follows:*

$$M_1 = 4(G_\infty + \sqrt{\epsilon})\Delta + C'(G_\infty + \sqrt{\epsilon})\log(2/\delta),$$

$$M_2 = \frac{4G_\infty^2(G_\infty + \sqrt{\epsilon})\epsilon^{-1/2}}{1 - \beta_1} + 4G_\infty(G_\infty + \sqrt{\epsilon}),$$

$$M_3 = \frac{4LG_\infty(G_\infty + \sqrt{\epsilon})}{\epsilon^{1/2}(1 - \beta_2)^{1/2}(1 - \beta_1/\beta_2^{1/2})}\left(1 + \frac{2\beta_1^2}{1 - \beta_1}\right),$$

*and $\Delta = f(\mathbf{x}_1) - \inf_{\mathbf{x}} f(\mathbf{x})$.*

**Remark 5.3** *Similar to the discussion in Remark 4.4, we can choose $\alpha = \Theta\big(d^{1/2}T^{1/4+s/2}\big)^{-1}$, to achieve an $O(d^{1/2}/T^{3/4-s/2} + d/T)$ convergence rate.*

We also have the following corollaries providing the high probability bounds for RMSProp and AdaGrad.

**Corollary 5.4 (A variant of RMSProp)** *Under the same conditions of Theorem 5.2, if $\alpha_t = \alpha \le \sigma^{-2}\epsilon/2$ and $\|\mathbf{g}_{1:T,i}\|_2 \le G_\infty T^s$ for $t = 1, \ldots, T, 0 \le s \le 1/2$, then for any $\delta > 0$, with probability at least $1 - \delta$, the iterates $\mathbf{x}_t$ of RMSProf satisfy that*

$$\frac{1}{T-1}\sum_{t=2}^{T}\|\nabla f(\mathbf{x}_t)\|_2^2 \le \frac{M_1}{T\alpha} + \frac{M_2 d}{T} + \frac{\alpha M_3 d}{T^{1/2-s}}, \tag{5.2}$$

*where $\{M_i\}_{i=1}^3$ are defined as follows:*

$$M_1 = 4(G_\infty + \sqrt{\epsilon})\Delta + C'(G_\infty + \sqrt{\epsilon})\log(2/\delta),$$

$$M_2 = 4G_\infty^2(G_\infty + \sqrt{\epsilon})\epsilon^{-1/2} + 4G_\infty^2,$$

$$M_3 = \frac{4LG_\infty(G_\infty + \sqrt{\epsilon})}{\epsilon^{1/2}(1 - \beta)^{1/2}},$$

*and $\Delta = f(\mathbf{x}_1) - \inf_{\mathbf{x}} f(\mathbf{x})$.*

**Corollary 5.5 (AdaGrad)** *Under the same conditions of Theorem 5.2, if $\alpha_t = \alpha \le \sigma^{-2}\epsilon/2$ and $\|\mathbf{g}_{1:T,i}\|_2 \le G_\infty T^s$ for $t = 1, \ldots, T, 0 \le s \le 1/2$, then for any $\delta > 0$, with probability at least $1 - \delta$, the iterates $\mathbf{x}_t$ of AdaGrad satisfy*

$$\frac{1}{T-1}\sum_{t=2}^{T}\|\nabla f(\mathbf{x}_t)\|_2^2 \le \frac{M_1}{T\alpha} + \frac{M_2 d}{T} + \frac{\alpha M_3 d}{T^{1/2-s}}, \tag{5.3}$$

*where $\{M_i\}_{i=1}^3$ are defined as follows:*

$$M_1 = (G_\infty + \sqrt{\epsilon})(4\Delta + C'\log(2/\delta)),$$

$$M_2 = (G_\infty + \sqrt{\epsilon})(4G_\infty^2\epsilon^{-1/2} + 4G_\infty),$$

$$M_3 = \frac{4LG_\infty(G_\infty + \sqrt{\epsilon})}{\epsilon^{1/2}},$$

*and $\Delta = f(\mathbf{x}_1) - \inf_{\mathbf{x}} f(\mathbf{x})$.*

# 6   Proof Sketch of the Main Results

In this section, we provide a proof sketch of Theorem 4.3 and Theorem 5.2, and the complete proofs as well as proofs for other corollaries and technical lemmas can be found in the supplemental materials. Compared with the analysis of standard stochastic gradient descent, the main difficulty of analyzing the convergence rate of adaptive gradient methods is caused by the stochastic momentum $\mathbf{m}_t$ and adaptive stochastic gradient $\widehat{\mathbf{V}}_t^{-1/2}\mathbf{g}_t$. To address this challenge, following Yang et al. (2016), we define an auxiliary sequence $\mathbf{z}_t$: let $\mathbf{x}_0 = \mathbf{x}_1$, and for each $t \geq 1$,

$$\mathbf{z}_t = \mathbf{x}_t + \frac{\beta_1}{1-\beta_1}(\mathbf{x}_t - \mathbf{x}_{t-1}) = \frac{1}{1-\beta_1}\mathbf{x}_t - \frac{\beta_1}{1-\beta_1}\mathbf{x}_{t-1}. \tag{6.1}$$

The following lemma shows that $\mathbf{z}_{t+1} - \mathbf{z}_t$ can be represented by $\mathbf{m}_t, \mathbf{g}_t$ and $\widehat{\mathbf{V}}_t^{-1/2}$. This indicates that by considering the sequence $\{\mathbf{z}_t\}$, it is possible to analyze algorithms which include stochastic momentum, such as AMSGrad.

**Lemma 6.1** *Let $\mathbf{z}_t$ be defined in* (6.1). *Then for $t \geq 2$, we have the following expression for $\mathbf{z}_{t+1} - \mathbf{z}_t$.*

$$\mathbf{z}_{t+1} - \mathbf{z}_t = \frac{\beta_1}{1-\beta_1}\Big[\mathbf{I} - \big(\alpha_t\widehat{\mathbf{V}}_t^{-1/2}\big)\big(\alpha_{t-1}\widehat{\mathbf{V}}_{t-1}^{-1/2}\big)^{-1}\Big](\mathbf{x}_{t-1} - \mathbf{x}_t) - \alpha_t\widehat{\mathbf{V}}_t^{-1/2}\mathbf{g}_t.$$

*We can also represent $\mathbf{z}_{t+1} - \mathbf{z}_t$ as the following:*

$$\mathbf{z}_{t+1} - \mathbf{z}_t = \frac{\beta_1}{1-\beta_1}\big(\alpha_{t-1}\widehat{\mathbf{V}}_{t-1}^{-1/2} - \alpha_t\widehat{\mathbf{V}}_t^{-1/2}\big)\mathbf{m}_{t-1} - \alpha_t\widehat{\mathbf{V}}_t^{-1/2}\mathbf{g}_t.$$

*For $t = 1$, we have $\mathbf{z}_2 - \mathbf{z}_1 = -\alpha_1\widehat{\mathbf{V}}_1^{-1/2}\mathbf{g}_1$.*

With Lemma 6.1, we have the following two lemmas giving upper bounds for $\|\mathbf{z}_{t+1} - \mathbf{z}_t\|_2$ and $\|\nabla f(\mathbf{z}_t) - \nabla f(\mathbf{x}_t)\|_2$ , which are useful for the proof of the main theorem.

**Lemma 6.2** *Let $\mathbf{z}_t$ be defined in* (6.1). *For $t \geq 2$, we have*

$$\|\mathbf{z}_{t+1} - \mathbf{z}_t\|_2 \leq \big\|\alpha\widehat{\mathbf{V}}_t^{-1/2}\mathbf{g}_t\big\|_2 + \frac{\beta_1}{1-\beta_1}\|\mathbf{x}_{t-1} - \mathbf{x}_t\|_2.$$

**Lemma 6.3** *Let $\mathbf{z}_t$ be defined in* (6.1). *For $t \geq 2$, we have*

$$\|\nabla f(\mathbf{z}_t) - \nabla f(\mathbf{x}_t)\|_2 \leq L\Big(\frac{\beta_1}{1-\beta_1}\Big) \cdot \|\mathbf{x}_t - \mathbf{x}_{t-1}\|_2.$$

We also need the following lemma to bound $\|\nabla f(\mathbf{x})\|_\infty, \|\widehat{\mathbf{v}}_t\|_\infty$ and $\|\mathbf{m}_t\|_\infty$. Basically, it shows that these quantities can be bounded by $G_\infty$.

**Lemma 6.4** *Let $\widehat{\mathbf{v}}_t$ and $\mathbf{m}_t$ be as defined in Algorithm 1. Then under Assumption 4.1, we have $\|\nabla f(\mathbf{x})\|_\infty \leq G_\infty$, $\|\widehat{\mathbf{v}}_t\|_\infty \leq G_\infty^2$ and $\|\mathbf{m}_t\|_\infty \leq G_\infty$.*

Lastly, we need the following lemma that provides upper bounds on $\|\widehat{\mathbf{V}}_t^{-1/2}\mathbf{m}_t\|_2$ and $\|\widehat{\mathbf{V}}_t^{-1/2}\mathbf{g}_t\|_2$. More specifically, it shows that we can bound $\|\widehat{\mathbf{V}}_t^{-1/2}\mathbf{m}_t\|_2$ and $\|\widehat{\mathbf{V}}_t^{-1/2}\mathbf{g}_t\|_2$ with $\sum_{i=1}^d \|\mathbf{g}_{1:T,i}\|_2$. The bound of $\big\|\widehat{\mathbf{V}}_t^{-1/2}\mathbf{m}_t\big\|_2^2$ is essential for us to obtain a tighter dependency in terms of $d$.

**Lemma 6.5** *Let $\beta_1, \beta_2$ be the weight parameters, $\alpha_t, t = 1, \ldots, T$ be the step sizes in Algorithm 1. We denote $\gamma = \beta_1/\beta_2^{1/2}$. Suppose that $\alpha_t = \alpha$ and $\gamma \leq 1$, then under Assumption 4.1, we have the following two results:*

$$\sum_{t=1}^T \alpha_t^2\big\|\widehat{\mathbf{V}}_t^{-1/2}\mathbf{m}_t\big\|_2^2 \leq \frac{T^{1/2}\alpha_t^2(1-\beta_1)}{2\epsilon^{1/2}(1-\beta_2)^{1/2}(1-\gamma)}\sum_{i=1}^d \|\mathbf{g}_{1:T,i}\|_2,$$

*and*

$$\sum_{t=1}^{T} \alpha_t^2 \big\| \widehat{\mathbf{V}}_t^{-1/2} \mathbf{g}_t \big\|_2^2 \leq \frac{T^{1/2} \alpha_t^2}{2\epsilon^{1/2}(1-\beta_2)^{1/2}(1-\gamma)} \sum_{i=1}^{d} \|\mathbf{g}_{1:T,i}\|_2.$$

With all lemmas provided above, now we are ready to provide the proof of Theorem 4.3.

**Proof** [Proof Sketch of Theorem 4.3] Since $f$ is $L$-smooth, we have:

$$f(\mathbf{z}_{t+1}) \leq f(\mathbf{z}_t) + \nabla f(\mathbf{z}_t)^\top (\mathbf{z}_{t+1} - \mathbf{z}_t) + \frac{L}{2}\|\mathbf{z}_{t+1} - \mathbf{z}_t\|_2^2$$

$$= f(\mathbf{z}_t) + \underbrace{\nabla f(\mathbf{x}_t)^\top (\mathbf{z}_{t+1} - \mathbf{z}_t)}_{I_1} + \underbrace{(\nabla f(\mathbf{z}_t) - \nabla f(\mathbf{x}_t))^\top (\mathbf{z}_{t+1} - \mathbf{z}_t)}_{I_2} + \underbrace{\frac{L}{2}\|\mathbf{z}_{t+1} - \mathbf{z}_t\|_2^2}_{I_3}. \tag{6.2}$$

In the following, we bound $I_1$, $I_2$ and $I_3$ separately.

**Bounding term $I_1$:** We can prove that when $t = 1$,

$$\nabla f(\mathbf{x}_1)^\top (\mathbf{z}_2 - \mathbf{z}_1) = -\nabla f(\mathbf{x}_1)^\top \alpha_1 \widehat{\mathbf{V}}_t^{-1/2} \mathbf{g}_1. \tag{6.3}$$

For $t \geq 2$, by Lemma 6.1, we can prove the following result:

$$\nabla f(\mathbf{x}_t)^\top (\mathbf{z}_{t+1} - \mathbf{z}_t) \leq \frac{1}{1-\beta_1} G_\infty^2 \Big( \big\| \alpha_{t-1} \widehat{\mathbf{v}}_{t-1}^{-1/2} \big\|_1 - \big\| \alpha_t \widehat{\mathbf{v}}_t^{-1/2} \big\|_1 \Big) - \nabla f(\mathbf{x}_t)^\top \alpha_{t-1} \widehat{\mathbf{V}}_{t-1}^{-1/2} \mathbf{g}_t. \tag{6.4}$$

**Bounding term $I_2$:** For $t \geq 1$, by Lemma 6.1 and Lemma 6.2, we can prove that

$$\big( \nabla f(\mathbf{z}_t) - \nabla f(\mathbf{x}_t) \big)^\top (\mathbf{z}_{t+1} - \mathbf{z}_t) \leq L \big\| \alpha_t \widehat{\mathbf{V}}_t^{-1/2} \mathbf{g}_t \big\|_2^2 + 2L \left( \frac{\beta_1}{1-\beta_1} \right)^2 \|\mathbf{x}_t - \mathbf{x}_{t-1}\|_2^2, \tag{6.5}$$

**Bounding term $I_3$:** For $t \geq 1$, by Lemma 6.1, we have

$$\frac{L}{2}\|\mathbf{z}_{t+1} - \mathbf{z}_t\|_2^2 \leq L \big\| \alpha_t \widehat{\mathbf{V}}_t^{-1/2} \mathbf{g}_t \big\|_2^2 + 2L \left( \frac{\beta_1}{1-\beta_1} \right)^2 \|\mathbf{x}_{t-1} - \mathbf{x}_t\|_2^2. \tag{6.6}$$

Now we get back to (6.2). We provide upper bounds of (6.2) for $t = 1$ and $t > 1$ separately. For $t = 1$, substituting (6.3), (6.5) and (6.6) into (6.2), taking expectation and rearranging terms, we have

$$\mathbb{E}[f(\mathbf{z}_2) - f(\mathbf{z}_1)] \leq \mathbb{E}[d\alpha_1 G_\infty + 2L \big\| \alpha_1 \widehat{\mathbf{V}}_1^{-1/2} \mathbf{g}_1 \big\|_2^2], \tag{6.7}$$

For $t \geq 2$, substituting (6.4), (6.5) and (6.6) into (6.2), taking expectation and rearranging terms, we have

$$\mathbb{E}\left[ f(\mathbf{z}_{t+1}) + \frac{G_\infty^2 \big\| \alpha_t \widehat{\mathbf{v}}_t^{-1/2} \big\|_1}{1-\beta_1} \right] - \mathbb{E}\left[ f(\mathbf{z}_t) + \frac{G_\infty^2 \big\| \alpha_{t-1} \widehat{\mathbf{v}}_{t-1}^{-1/2} \big\|_1}{1-\beta_1} \right]$$

$$\leq \mathbb{E}\left[ -\alpha_{t-1} \big\| \nabla f(\mathbf{x}_t) \big\|_2^2 (G_\infty + \sqrt{\epsilon})^{-1} + 2L \big\| \alpha_t \widehat{\mathbf{V}}_t^{-1/2} \mathbf{g}_t \big\|_2^2 + 4L \left( \frac{\beta_1}{1-\beta_1} \right)^2 \big\| \alpha_{t-1} \widehat{\mathbf{V}}_{t-1}^{-1/2} \mathbf{m}_{t-1} \big\|_2^2 \right], \tag{6.8}$$

where the inequality holds due to the fact $\nabla f(\mathbf{x}_t)^\top \widehat{\mathbf{V}}_{t-1}^{-1/2} \nabla f(\mathbf{x}_t) \geq (G_\infty + \sqrt{\epsilon})^{-1} \|\nabla f(\mathbf{x}_t)\|_2^2$ by Lemma 6.4. We now telescope (6.8) for $t = 2$ to $T$, and add it with (6.7). Rearranging it, we have

$$(G_\infty + \sqrt{\epsilon})^{-1} \sum_{t=2}^{T} \alpha_{t-1} \mathbb{E} \big\| \nabla f(\mathbf{x}_t) \big\|_2^2$$

$$\leq \mathbb{E}\left[ \Delta + \frac{G_\infty^2 \alpha_1 \epsilon^{-1/2} d}{1-\beta_1} + d\alpha_1 G_\infty \right] + 2L \sum_{t=1}^{T} \mathbb{E} \big\| \alpha_t \widehat{\mathbf{V}}_t^{-1/2} \mathbf{g}_t \big\|_2^2 + 4L \left( \frac{\beta_1}{1-\beta_1} \right)^2 \sum_{t=1}^{T} \mathbb{E}\left[ \big\| \alpha_t \widehat{\mathbf{V}}_t^{-1/2} \mathbf{m}_t \big\|_2^2 \right]. \tag{6.9}$$

By using Lemma 6.5, we can further bound $\sum_{t=1}^{T} \mathbb{E}\|\alpha_t \widehat{\mathbf{V}}_t^{-1/2} \mathbf{g}_t\|_2^2$ and $\sum_{t=1}^{T} \mathbb{E}\|\alpha_t \widehat{\mathbf{V}}_t^{-1/2} \mathbf{m}_t\|_2^2$ in (6.9) with $\sum_{i=1}^{d} \|\mathbf{g}_{1:T,i}\|_2$, which turns out to be

$$\mathbb{E}\|\nabla f(\mathbf{x}_{\mathrm{out}})\|_2^2 \leq \frac{1}{T\alpha} 2(G_\infty + \sqrt{\epsilon})\Delta + \frac{2}{T}\left(\frac{G_\infty^2(G_\infty + \sqrt{\epsilon})\epsilon^{-1/2}d}{1-\beta_1} + dG_\infty(G_\infty + \sqrt{\epsilon})\right)$$
$$+ \frac{2(G_\infty + \sqrt{\epsilon})L\alpha}{T^{1/2}\epsilon^{1/2}(1-\gamma)(1-\beta_2)^{1/2}} \mathbb{E}\left(\sum_{i=1}^{d} \|\mathbf{g}_{1:T,i}\|_2\right) \cdot \left(1 + 2(1-\beta_1)\left(\frac{\beta_1}{1-\beta_1}\right)^2\right), \quad (6.10)$$

Finally, rearranging (6.10), and adopting the theorem condition that $\|\mathbf{g}_{1:T,i}\|_2 \leq G_\infty T^s$, we obtain

$$\mathbb{E}\|\nabla f(\mathbf{x}_{\mathrm{out}})\|_2^2 \leq \frac{M_1}{T\alpha} + \frac{M_2 d}{T} + \frac{\alpha M_3 d}{T^{1/2-s}},$$

where $\{M_i\}_{i=1}^3$ are defined in Theorem 4.3. This completes the proof. ■

**Remark 6.6** *We highlight here why we can achieve a tighter dimension dependency ($d/\sqrt{T}$ v.s. $\sqrt{d}/\sqrt{T}$) as compared with Défossez et al. (2020). Both our analysis and the one in Défossez et al. (2020) required to upper bound the gradient norm $\|\nabla f(\mathbf{x}_{out})\|_2^2$ by the stochastic gradients $\mathbf{g}_t$ and momentum $\mathbf{m}_t$ (see our (6.9) and (A.19) in Défossez et al. (2020). However, Défossez et al. (2020) bounded $\mathbf{m}_t$ and $\mathbf{g}_t$ separately as suggested by (A.20) in Défossez et al. (2020), and they obtained a better bound for $\mathbf{m}_t$, which depends on $\alpha^2$, and a worse bound for $\mathbf{g}_t$, which has an $\alpha^0$ dependency. Thus, the final bound in their result suffers from an $\alpha^2 d + d = O(d)$ dependency (see the second and third term in (A.54) in Défossez et al. (2020). To compare with, we bound both $m_t$ and $g_t$ by $\sum_{i=1}^{d} \|\mathbf{g}_{1:T,i}\|_2$ uniformly by using Lemma 6.5 which makes our final bound only has an $\alpha^1 d$ dependency (see the third term in (6.10)). Therefore, by optimizing $\alpha$, our final bound only depends on $\sqrt{d}$ rather than $d$.*

We then show the proof sketch for high probability result, i.e, Theorem 4.3.

**Proof** [Proof Sketch of Theorem 5.2] Following the same procedure as in the proof for Theorem 4.3 until (6.6). For $t = 1$, substituting (6.3), (6.5) and (6.6) into (6.2), rearranging terms, we have

$$f(\mathbf{z}_2) - f(\mathbf{z}_1) \leq d\alpha_1 G_\infty + 2L\|\alpha_1 \widehat{\mathbf{V}}_1^{-1/2} \mathbf{g}_1\|_2^2, \quad (6.11)$$

For $t \geq 2$, substituting (6.4), (6.5) and (6.6) into (6.2), rearranging terms, we have

$$f(\mathbf{z}_{t+1}) + \frac{G_\infty^2 \|\alpha_t \widehat{\mathbf{V}}_t^{-1/2}\|_{1,1}}{1-\beta_1} - \left(f(\mathbf{z}_t) + \frac{G_\infty^2 \|\alpha_{t-1} \widehat{\mathbf{V}}_{t-1}^{-1/2}\|_{1,1}}{1-\beta_1}\right)$$
$$\leq -\nabla f(\mathbf{x}_t)^\top \alpha_{t-1} \widehat{\mathbf{V}}_{t-1}^{-1/2} \mathbf{g}_t + 2L\|\alpha_t \widehat{\mathbf{V}}_t^{-1/2} \mathbf{g}_t\|_2^2 + 4L\left(\frac{\beta_1}{1-\beta_1}\right)^2 \|\alpha_{t-1} \widehat{\mathbf{V}}_{t-1}^{-1/2} \mathbf{m}_{t-1}\|_2^2. \quad (6.12)$$

We now telescope (6.12) for $t = 2$ to $T$ and add it with (6.11). Rearranging it, we have

$$\sum_{t=2}^{T} \alpha_{t-1} \nabla f(\mathbf{x}_t)^\top \widehat{\mathbf{V}}_{t-1}^{-1/2} \mathbf{g}_t$$
$$\leq \Delta + \frac{G_\infty^2 \alpha_1 \epsilon^{-1/2} d}{1-\beta_1} + d\alpha_1 G_\infty + 2L\sum_{t=1}^{T} \|\alpha_t \widehat{\mathbf{V}}_t^{-1/2} \mathbf{g}_t\|_2^2 + 4L\left(\frac{\beta_1}{1-\beta_1}\right)^2 \sum_{t=1}^{T} \|\alpha_t \widehat{\mathbf{V}}_t^{-1/2} \mathbf{m}_t\|_2^2. \quad (6.13)$$

Now consider the filtration $\mathcal{F}_t = \sigma(\xi_1, \ldots, \xi_t)$. Since $\mathbf{x}_t$ and $\widehat{\mathbf{V}}_{t-1}^{-1/2}$ only depend on $\xi_1, \ldots, \xi_{t-1}$, by Assumption 5.1 and an martingale concentration argument, we obtain

$$\left|\sum_{t=2}^{T} \alpha_{t-1} \nabla f(\mathbf{x}_t)^\top \widehat{\mathbf{V}}_{t-1}^{-1/2} \mathbf{g}_t - \sum_{t=2}^{T} \alpha_{t-1} \nabla f(\mathbf{x}_t)^\top \widehat{\mathbf{V}}_{t-1}^{-1/2} \nabla f(\mathbf{x}_t)\right|$$
$$\leq \epsilon^{-1}\sigma^2 \sum_{t=2}^{T} \alpha_{t-1}^2 \|\nabla f(\mathbf{x}_t)\|_2^2 + C\log(2/\delta), \quad (6.14)$$

By using Lemma 6.5 and substituting (6.14) into (6.13), we have

$$\sum_{t=2}^{T} \alpha_{t-1} \nabla f(\mathbf{x}_t)^\top \widehat{\mathbf{V}}_{t-1}^{-1/2} \nabla f(\mathbf{x}_t)$$

$$\leq \Delta + \frac{G_\infty^2 \alpha_1 \epsilon^{-1/2} d}{1 - \beta_1} + d\alpha_1 G_\infty + \epsilon^{-1} \sigma^2 \sum_{t=2}^{T} \alpha_{t-1}^2 \|\nabla f(\mathbf{x}_t)\|_2^2 + \frac{L T^{1/2} \alpha_t^2}{\epsilon^{1/2}(1 - \beta_2)^{1/2}(1 - \gamma)} \sum_{i=1}^{d} \|\mathbf{g}_{1:T,i}\|_2$$

$$+ C \log(2/\delta) + \left(\frac{\beta_1}{1 - \beta_1}\right)^2 \frac{2 L T^{1/2} \alpha_t^2 (1 - \beta_1)}{\epsilon^{1/2}(1 - \beta_2)^{1/2}(1 - \gamma)} \sum_{i=1}^{d} \|\mathbf{g}_{1:T,i}\|_2.$$

Moreover, by Lemma 6.4, we have $\nabla f(\mathbf{x}_t)^\top \widehat{\mathbf{V}}_{t-1}^{-1/2} \nabla f(\mathbf{x}_t) \geq (G_\infty + \sqrt{\epsilon})^{-1} \|\nabla f(\mathbf{x}_t)\|_2^2$, and therefore by choosing $\alpha_t = \alpha \leq \sigma^{-2}\epsilon/2$ and rearranging terms, we have

$$\frac{1}{T-1} \sum_{t=2}^{T} \|\nabla f(\mathbf{x}_t)\|_2^2 \leq \frac{4(G_\infty + \sqrt{\epsilon})}{T\alpha} \cdot \Delta + \frac{4 G_\infty^2 (G_\infty + \sqrt{\epsilon})\epsilon^{-1/2}}{1 - \beta_1} \cdot \frac{d}{T} + 4 G_\infty (G_\infty + \sqrt{\epsilon}) \cdot \frac{d}{T}$$

$$+ \frac{4(G_\infty + \sqrt{\epsilon})L\alpha}{\epsilon^{1/2}(1 - \beta_2)^{1/2}(1 - \gamma)T^{1/2}} \sum_{i=1}^{d} \|\mathbf{g}_{1:T,i}\|_2 + \frac{C'(G_\infty + \sqrt{\epsilon})\log(2/\delta)}{T\alpha}$$

$$+ \left(\frac{\beta_1}{1 - \beta_1}\right)^2 \frac{8(G_\infty + \sqrt{\epsilon})L\alpha(1 - \beta_1)}{\epsilon^{1/2}(1 - \beta_2)^{1/2}(1 - \gamma)T^{1/2}} \sum_{i=1}^{d} \|\mathbf{g}_{1:T,i}\|_2, \tag{6.15}$$

where $C'$ is an absolute constant. Finally, rearranging (6.15) and adopting the condition $\|\mathbf{g}_{1:T,i}\|_2 \leq G_\infty T^s$ gives

$$\frac{1}{T-1} \sum_{t=2}^{T} \|\nabla f(\mathbf{x}_t)\|_2^2 \leq \frac{M_1}{T\alpha} + \frac{M_2 d}{T} + \frac{\alpha M_3 d}{T^{1/2-s}},$$

where $\{M_i\}_{i=1}^3$ are defined in Theorem 5.2. This completes the proof. ∎

# 7 Conclusion

In this paper, we provided a fine-grained analysis of a general class of adaptive gradient methods, and proved their convergence rates for smooth nonconvex optimization. Our results provide faster convergence rates of AMSGrad and the corrected version of RMSProp as well as AdaGrad for smooth nonconvex optimization compared with previous works. In addition, we also prove high probability bounds on the convergence rates of AMSGrad and RMSProp as well as AdaGrad, which have not been established before.

### Acknowledgments

We thank Yiqi Tang for his valuable discussions and preparation of this work. We thank the anonymous reviewers for their helpful comments. This research was sponsored in part by the National Science Foundation CAREER Award IIS-1906169, BIGDATA IIS-1855099 and IIS-2008981. We also thank AWS for providing cloud computing credits associated with the NSF BIGDATA award. The views and conclusions contained in this paper are those of the authors and should not be interpreted as representing any funding agencies.

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

## Appendix

## A  Proof of the Main Theory

Here we provide the detailed proof of the main theorem.

### A.1  Proof of Theorem 4.3

Let $\mathbf{x}_0 = \mathbf{x}_1$. To prove Theorem 4.3, we need the following lemmas:

**Lemma A.1 (Restatement of Lemma 6.4)** *Let $\widehat{\mathbf{v}}_t$ and $\mathbf{m}_t$ be as defined in Algorithm 1. Then under Assumption 4.1, we have $\|\nabla f(\mathbf{x})\|_\infty \le G_\infty$, $\|\widehat{\mathbf{v}}_t\|_\infty \le G_\infty^2$ and $\|\mathbf{m}_t\|_\infty \le G_\infty$.*

**Lemma A.2 (Generalized version of Lemma 6.5)** *Let $\beta_1, \beta_2, \beta_1', \beta_2'$ be the weight parameters such that*

$$\mathbf{m}_t = \beta_1 \mathbf{m}_{t-1} + (1 - \beta_1')\mathbf{g}_t,$$
$$\mathbf{v}_t = \beta_2 \mathbf{v}_{t-1} + (1 - \beta_2')\mathbf{g}_t^2,$$

*$\alpha_t$, $t = 1, \ldots, T$ be the step sizes. We denote $\gamma = \beta_1/\beta_2^{1/2}$. Suppose that $\alpha_t = \alpha$ and $\gamma \le 1$, then under Assumption 4.1, we have the following two results:*

$$\sum_{t=1}^T \alpha_t^2 \big\|\widehat{\mathbf{V}}_t^{-1/2}\mathbf{m}_t\big\|_2^2 \le \frac{T^{1/2}\alpha_t^2(1 - \beta_1')}{2\epsilon^{1/2}(1 - \beta_2')^{1/2}(1 - \gamma)} \sum_{i=1}^d \|\mathbf{g}_{1:T,i}\|_2,$$

*and*

$$\sum_{t=1}^T \alpha_t^2 \big\|\widehat{\mathbf{V}}_t^{-1/2}\mathbf{g}_t\big\|_2^2 \le \frac{T^{1/2}\alpha_t^2}{2\epsilon^{1/2}(1 - \beta_2')^{1/2}(1 - \gamma)} \sum_{i=1}^d \|\mathbf{g}_{1:T,i}\|_2.$$

Note that Lemma A.2 is general and applicable to various algorithms. Specifically, set $\beta_1' = \beta_1$ and $\beta_2' = \beta_2$, we recover the case in Algorithm 1. Further set $\beta_1 = 0$ we recover the case in Algorithm 2. Set $\beta_1' = \beta_1 = 0$ and $\beta_2 = 1, \beta_2' = 0$ we recover the case in Algorithm 3.

To deal with stochastic momentum $\mathbf{m}_t$ and stochastic weight $\widehat{\mathbf{V}}_t^{-1/2}$, following Yang et al. (2016), we define an auxiliary sequence $\mathbf{z}_t$ as follows: let $\mathbf{x}_0 = \mathbf{x}_1$, and for each $t \ge 1$,

$$\mathbf{z}_t = \mathbf{x}_t + \frac{\beta_1}{1 - \beta_1}(\mathbf{x}_t - \mathbf{x}_{t-1}) = \frac{1}{1 - \beta_1}\mathbf{x}_t - \frac{\beta_1}{1 - \beta_1}\mathbf{x}_{t-1}. \tag{A.1}$$

Lemma A.3 shows that $\mathbf{z}_{t+1} - \mathbf{z}_t$ can be represented in two different ways.

**Lemma A.3 (Restatement of Lemma 6.1)** *Let $\mathbf{z}_t$ be defined in* (A.1). *For $t \ge 2$, we have*

$$\mathbf{z}_{t+1} - \mathbf{z}_t = \frac{\beta_1}{1 - \beta_1}\Big[\mathbf{I} - \big(\alpha_t\widehat{\mathbf{V}}_t^{-1/2}\big)\big(\alpha_{t-1}\widehat{\mathbf{V}}_{t-1}^{-1/2}\big)^{-1}\Big](\mathbf{x}_{t-1} - \mathbf{x}_t) - \alpha_t\widehat{\mathbf{V}}_t^{-1/2}\mathbf{g}_t. \tag{A.2}$$

*and*

$$\mathbf{z}_{t+1} - \mathbf{z}_t = \frac{\beta_1}{1 - \beta_1}\big(\alpha_{t-1}\widehat{\mathbf{V}}_{t-1}^{-1/2} - \alpha_t\widehat{\mathbf{V}}_t^{-1/2}\big)\mathbf{m}_{t-1} - \alpha_t\widehat{\mathbf{V}}_t^{-1/2}\mathbf{g}_t. \tag{A.3}$$

*For $t = 1$, we have*

$$\mathbf{z}_2 - \mathbf{z}_1 = -\alpha_1\widehat{\mathbf{V}}_1^{-1/2}\mathbf{g}_1. \tag{A.4}$$

By Lemma A.3, we connect $\mathbf{z}_{t+1} - \mathbf{z}_t$ with $\mathbf{x}_{t+1} - \mathbf{x}_t$ and $\alpha_t\widehat{\mathbf{V}}_t^{-1/2}\mathbf{g}_t$. The following two lemmas give bounds on $\|\mathbf{z}_{t+1} - \mathbf{z}_t\|_2$ and $\|\nabla f(\mathbf{z}_t) - \nabla f(\mathbf{x}_t)\|_2$, which play important roles in our proof.

**Lemma A.4 (Restatement of Lemma 6.2)** *Let $\mathbf{z}_t$ be defined in* (A.1). *For $t \geq 2$, we have*

$$\|\mathbf{z}_{t+1} - \mathbf{z}_t\|_2 \leq \left\|\alpha\widehat{\mathbf{V}}_t^{-1/2}\mathbf{g}_t\right\|_2 + \frac{\beta_1}{1 - \beta_1}\|\mathbf{x}_{t-1} - \mathbf{x}_t\|_2.$$

**Lemma A.5 (Restatement of Lemma 6.3)** *Let $\mathbf{z}_t$ be defined in* (A.1). *For $t \geq 2$, we have*

$$\|\nabla f(\mathbf{z}_t) - \nabla f(\mathbf{x}_t)\|_2 \leq L\left(\frac{\beta_1}{1 - \beta_1}\right) \cdot \|\mathbf{x}_t - \mathbf{x}_{t-1}\|_2.$$

We present the following lemma which upper bounds the difference $f(\mathbf{z}_{t+1}) - f(\mathbf{z}_t)$.

**Lemma A.6** *For $t = 1$, we have*

$$f(\mathbf{z}_2) - f(\mathbf{z}_1) \leq d\alpha_1 G_\infty + 2L\left\|\alpha_1\widehat{\mathbf{V}}_1^{-1/2}\mathbf{g}_1\right\|_2^2.$$

*For $t \geq 2$, we have*

$$
\begin{aligned}
f(\mathbf{z}_{t+1}) + &\frac{G_\infty^2\left\|\alpha_t\widehat{\mathbf{V}}_t^{-1/2}\right\|_{1,1}}{1 - \beta_1} - \left(f(\mathbf{z}_t) + \frac{G_\infty^2\left\|\alpha_{t-1}\widehat{\mathbf{V}}_{t-1}^{-1/2}\right\|_{1,1}}{1 - \beta_1}\right) \\
&\leq -\nabla f(\mathbf{x}_t)^\top\alpha_{t-1}\widehat{\mathbf{V}}_{t-1}^{-1/2}\mathbf{g}_t + 2L\left\|\alpha_t\widehat{\mathbf{V}}_t^{-1/2}\mathbf{g}_t\right\|_2^2 + 4L\left(\frac{\beta_1}{1 - \beta_1}\right)^2\|\mathbf{x}_t - \mathbf{x}_{t-1}\|_2^2,
\end{aligned}
$$

Now we are ready to prove Theorem 4.3.

**Proof** [Proof of Theorem 4.3]

By Lemma A.6, for $t = 1$, we have

$$\mathbb{E}[f(\mathbf{z}_2) - f(\mathbf{z}_1)] \leq \mathbb{E}[d\alpha_1 G_\infty + 2L\left\|\alpha_1\widehat{\mathbf{V}}_1^{-1/2}\mathbf{g}_1\right\|_2^2]. \tag{A.5}$$

For $t \geq 2$, we have

$$
\begin{aligned}
\mathbb{E}\Bigg[f(\mathbf{z}_{t+1}) + &\frac{G_\infty^2\left\|\alpha_t\widehat{\mathbf{V}}_t^{-1/2}\right\|_{1,1}}{1 - \beta_1} - \left(f(\mathbf{z}_t) + \frac{G_\infty^2\left\|\alpha_{t-1}\widehat{\mathbf{V}}_{t-1}^{-1/2}\right\|_{1,1}}{1 - \beta_1}\right)\Bigg] \\
&\leq \mathbb{E}\left[-\nabla f(\mathbf{x}_t)^\top\alpha_{t-1}\widehat{\mathbf{V}}_{t-1}^{-1/2}\mathbf{g}_t + 2L\left\|\alpha_t\widehat{\mathbf{V}}_t^{-1/2}\mathbf{g}_t\right\|_2^2 + 4L\left(\frac{\beta_1}{1 - \beta_1}\right)^2\|\mathbf{x}_t - \mathbf{x}_{t-1}\|_2^2\right] \\
&= \mathbb{E}\left[-\nabla f(\mathbf{x}_t)^\top\alpha_{t-1}\widehat{\mathbf{V}}_{t-1}^{-1/2}\nabla f(\mathbf{x}_t) + 2L\left\|\alpha_t\widehat{\mathbf{V}}_t^{-1/2}\mathbf{g}_t\right\|_2^2 + 4L\left(\frac{\beta_1}{1 - \beta_1}\right)^2\left\|\alpha_{t-1}\widehat{\mathbf{V}}_{t-1}^{-1/2}\mathbf{m}_{t-1}\right\|_2^2\right] \\
&\leq \mathbb{E}\left[-\alpha_{t-1}\|\nabla f(\mathbf{x}_t)\|_2^2(G_\infty + \sqrt{\epsilon})^{-1} + 2L\left\|\alpha_t\widehat{\mathbf{V}}_t^{-1/2}\mathbf{g}_t\right\|_2^2 + 4L\left(\frac{\beta_1}{1 - \beta_1}\right)^2\left\|\alpha_{t-1}\widehat{\mathbf{V}}_{t-1}^{-1/2}\mathbf{m}_{t-1}\right\|_2^2\right], \tag{A.6}
\end{aligned}
$$

where the equality holds because $\mathbb{E}[\mathbf{g}_t] = \nabla f(\mathbf{x}_t)$ conditioned on $\nabla f(\mathbf{x}_t)$ and $\widehat{\mathbf{V}}_{t-1}^{-1/2}$, the second inequality holds because of Lemma A.1. Telescoping (A.6) for $t = 2$ to $T$ and adding with (B.15), we have

$$
(G_\infty + \sqrt{\epsilon})^{-1} \sum_{t=2}^{T} \alpha_{t-1} \mathbb{E}\big\|\nabla f(\mathbf{x}_t)\big\|_2^2
$$

$$
\leq \mathbb{E}\bigg[f(\mathbf{z}_1) + \frac{G_\infty^2 \big\|\alpha_1 \widehat{\mathbf{V}}_1^{-1/2}\big\|_{1,1}}{1 - \beta_1} + d\alpha_1 G_\infty - \bigg(f(\mathbf{z}_{T+1}) + \frac{G_\infty^2 \big\|\alpha_T \widehat{\mathbf{v}}_T^{-1/2}\big\|_1}{1 - \beta_1}\bigg)\bigg]
$$

$$
+ 2L \sum_{t=1}^{T} \mathbb{E}\big\|\alpha_t \widehat{\mathbf{V}}_t^{-1/2} \mathbf{g}_t\big\|_2^2 + 4L\bigg(\frac{\beta_1}{1 - \beta_1}\bigg)^2 \sum_{t=2}^{T} \mathbb{E}\Big[\big\|\alpha_{t-1} \widehat{\mathbf{V}}_{t-1}^{-1/2} \mathbf{m}_{t-1}\big\|_2^2\Big]
$$

$$
\leq \mathbb{E}\bigg[\Delta + \frac{G_\infty^2 \alpha_1 \epsilon^{-1/2} d}{1 - \beta_1} + d\alpha_1 G_\infty\bigg] + 2L \sum_{t=1}^{T} \mathbb{E}\big\|\alpha_t \widehat{\mathbf{V}}_t^{-1/2} \mathbf{g}_t\big\|_2^2
$$

$$
+ 4L\bigg(\frac{\beta_1}{1 - \beta_1}\bigg)^2 \sum_{t=1}^{T} \mathbb{E}\Big[\big\|\alpha_t \widehat{\mathbf{V}}_t^{-1/2} \mathbf{m}_t\big\|_2^2\Big]. \tag{A.7}
$$

By Lemma A.2, we have

$$
\sum_{t=1}^{T} \alpha_t^2 \mathbb{E}\Big[\big\|\widehat{\mathbf{V}}_t^{-1/2} \mathbf{m}_t\big\|_2^2\Big] \leq \frac{T^{1/2} \alpha_t^2 (1 - \beta_1)}{2\epsilon^{1/2} (1 - \beta_2)^{1/2} (1 - \gamma)} \mathbb{E}\bigg(\sum_{i=1}^{d} \|\mathbf{g}_{1:T,i}\|_2\bigg), \tag{A.8}
$$

where $\gamma = \beta_1 / \beta_2^{1/2}$. We also have

$$
\sum_{t=1}^{T} \alpha_t^2 \mathbb{E}\Big[\big\|\widehat{\mathbf{V}}_t^{-1/2} \mathbf{g}_t\big\|_2^2\Big] \leq \frac{T^{1/2} \alpha_t^2}{2\epsilon^{1/2} (1 - \beta_2)^{1/2} (1 - \gamma)} \mathbb{E}\bigg(\sum_{i=1}^{d} \|\mathbf{g}_{1:T,i}\|_2\bigg). \tag{A.9}
$$

Substituting (A.8) and (A.9) into (A.7), and rearranging (A.7), we have

$$
\mathbb{E}\|\nabla f(\mathbf{x}_{\text{out}})\|_2^2 = \frac{1}{\sum_{t=2}^{T} \alpha_{t-1}} \sum_{t=2}^{T} \alpha_{t-1} \mathbb{E}\big\|\nabla f(\mathbf{x}_t)\big\|_2^2
$$

$$
\leq \frac{(G_\infty + \sqrt{\epsilon})}{\sum_{t=2}^{T} \alpha_{t-1}} \mathbb{E}\bigg[\Delta + \frac{G_\infty^2 \alpha_1 \epsilon^{-1/2} d}{1 - \beta_1} + d\alpha_1 G_\infty\bigg]
$$

$$
+ \frac{2L(G_\infty + \sqrt{\epsilon})}{\sum_{t=2}^{T} \alpha_{t-1}} \cdot \frac{T^{1/2} \alpha_t^2}{2\epsilon^{1/2} (1 - \beta_2)^{1/2} (1 - \gamma)} \cdot \mathbb{E}\bigg(\sum_{i=1}^{d} \|\mathbf{g}_{1:T,i}\|_2\bigg)
$$

$$
+ \frac{4L(G_\infty + \sqrt{\epsilon})}{\sum_{t=2}^{T} \alpha_{t-1}} \bigg(\frac{\beta_1}{1 - \beta_1}\bigg)^2 \frac{T^{1/2} \alpha_t^2 (1 - \beta_1)}{2\epsilon^{1/2} (1 - \beta_2)^{1/2} (1 - \gamma)} \cdot \mathbb{E}\bigg(\sum_{i=1}^{d} \|\mathbf{g}_{1:T,i}\|_2\bigg)
$$

$$
\leq \frac{1}{T\alpha} 2(G_\infty + \sqrt{\epsilon})\Delta + \frac{2}{T}\bigg(\frac{G_\infty^2 (G_\infty + \sqrt{\epsilon}) \epsilon^{-1/2} d}{1 - \beta_1} + dG_\infty (G_\infty + \sqrt{\epsilon})\bigg)
$$

$$
+ \frac{2(G_\infty + \sqrt{\epsilon})L\alpha}{T^{1/2} \epsilon^{1/2} (1 - \gamma)(1 - \beta_2)^{1/2}} \mathbb{E}\bigg(\sum_{i=1}^{d} \|\mathbf{g}_{1:T,i}\|_2\bigg) \cdot \bigg(1 + 2(1 - \beta_1)\bigg(\frac{\beta_1}{1 - \beta_1}\bigg)^2\bigg), \tag{A.10}
$$

where the second inequality holds because $\alpha_t = \alpha$. Rearranging (A.10), and note that in the theorem condition we have $\|\mathbf{g}_{1:T,i}\|_2 \leq G_\infty T^s$, we obtain

$$
\mathbb{E}\|\nabla f(\mathbf{x}_{\text{out}})\|_2^2 \leq \frac{M_1}{T\alpha} + \frac{M_2 d}{T} + \frac{\alpha M_3 d}{T^{1/2-s}},
$$

where $\{M_i\}_{i=1}^{3}$ are defined in Theorem 4.3. This completes the proof. ∎

## A.2 Proof of Corollary 4.5

**Proof** [Proof of Corollary 4.5] Following the proof for Theorem 4.3, setting $\beta_1' = \beta_1 = 0$ and $\beta_2' = \beta_2 = \beta$ in Lemma A.2 we get the conclusion. ∎

## A.3 Proof of Corollary 4.6

**Proof** [Proof of Corollary 4.6] Following the proof for Theorem 4.3, setting $\beta_1' = \beta_1 = 0$, $\beta_2 = 1$ and $\beta_2' = 0$ in Lemma A.2 we get the conclusion. ∎

## A.4 Proof of Theorem 5.2

**Proof** [Proof of Theorem 5.2]

By Lemma A.6, for $t = 1$, we have

$$f(\mathbf{z}_2) - f(\mathbf{z}_1) \leq d\alpha_1 G_\infty + 2L\big\|\alpha_1 \widehat{\mathbf{V}}_1^{-1/2} \mathbf{g}_1\big\|_2^2, \tag{A.11}$$

For $t \geq 2$, we have

$$
\begin{aligned}
f(\mathbf{z}_{t+1}) &+ \frac{G_\infty^2 \big\|\alpha_t \widehat{\mathbf{V}}_t^{-1/2}\big\|_{1,1}}{1 - \beta_1} - \left( f(\mathbf{z}_t) + \frac{G_\infty^2 \big\|\alpha_{t-1} \widehat{\mathbf{V}}_{t-1}^{-1/2}\big\|_{1,1}}{1 - \beta_1} \right) \\
&\leq -\nabla f(\mathbf{x}_t)^\top \alpha_{t-1} \widehat{\mathbf{V}}_{t-1}^{-1/2} \mathbf{g}_t + 2L\big\|\alpha_t \widehat{\mathbf{V}}_t^{-1/2} \mathbf{g}_t\big\|_2^2 + 4L\left(\frac{\beta_1}{1 - \beta_1}\right)^2 \|\mathbf{x}_t - \mathbf{x}_{t-1}\|_2^2 \\
&= -\nabla f(\mathbf{x}_t)^\top \alpha_{t-1} \widehat{\mathbf{V}}_{t-1}^{-1/2} \mathbf{g}_t + 2L\big\|\alpha_t \widehat{\mathbf{V}}_t^{-1/2} \mathbf{g}_t\big\|_2^2 + 4L\left(\frac{\beta_1}{1 - \beta_1}\right)^2 \big\|\alpha_{t-1} \widehat{\mathbf{V}}_{t-1}^{-1/2} \mathbf{m}_{t-1}\big\|_2^2.
\end{aligned}
\tag{A.12}
$$

Telescoping (A.12) for $t = 2$ to $T$ and adding (A.11), we have

$$
\begin{aligned}
\sum_{t=2}^T \alpha_{t-1} \nabla f(\mathbf{x}_t)^\top \widehat{\mathbf{V}}_{t-1}^{-1/2} \mathbf{g}_t &\leq f(\mathbf{z}_1) + \frac{G_\infty^2 \big\|\alpha_1 \widehat{\mathbf{V}}_1^{-1/2}\big\|_{1,1}}{1 - \beta_1} + d\alpha_1 G_\infty - \left( f(\mathbf{z}_{T+1}) + \frac{G_\infty^2 \big\|\alpha_T \widehat{\mathbf{v}}_T^{-1/2}\big\|_1}{1 - \beta_1} \right) \\
&\quad + 2L \sum_{t=1}^T \big\|\alpha_t \widehat{\mathbf{V}}_t^{-1/2} \mathbf{g}_t\big\|_2^2 + 4L\left(\frac{\beta_1}{1 - \beta_1}\right)^2 \sum_{t=2}^T \big\|\alpha_{t-1} \widehat{\mathbf{V}}_{t-1}^{-1/2} \mathbf{m}_{t-1}\big\|_2^2 \\
&\leq \Delta + \frac{G_\infty^2 \alpha_1 \epsilon^{-1/2} d}{1 - \beta_1} + d\alpha_1 G_\infty + 2L \sum_{t=1}^T \big\|\alpha_t \widehat{\mathbf{V}}_t^{-1/2} \mathbf{g}_t\big\|_2^2 \\
&\quad + 4L\left(\frac{\beta_1}{1 - \beta_1}\right)^2 \sum_{t=1}^T \big\|\alpha_t \widehat{\mathbf{V}}_t^{-1/2} \mathbf{m}_t\big\|_2^2.
\end{aligned}
\tag{A.13}
$$

By Lemma A.2, we have

$$\sum_{t=1}^T \alpha_t^2 [\|\widehat{\mathbf{V}}_t^{-1/2} \mathbf{m}_t\|_2^2 \leq \frac{T^{1/2} \alpha_t^2 (1 - \beta_1)}{2\epsilon^{1/2}(1 - \beta_2)^{1/2}(1 - \gamma)} \sum_{i=1}^d \|\mathbf{g}_{1:T,i}\|_2, \tag{A.14}$$

where $\gamma = \beta_1 / \beta_2^{1/2}$. We also have

$$\sum_{t=1}^T \alpha_t^2 \|\widehat{\mathbf{V}}_t^{-1/2} \mathbf{g}_t\|_2^2 \leq \frac{T^{1/2} \alpha_t^2}{2\epsilon^{1/2}(1 - \beta_2)^{1/2}(1 - \gamma)} \sum_{i=1}^d \|\mathbf{g}_{1:T,i}\|_2. \tag{A.15}$$

Moreover, consider the filtration $\mathcal{F}_t = \sigma(\xi_1, \ldots, \xi_t)$. Since $\mathbf{x}_t$ and $\widehat{\mathbf{V}}_{t-1}^{-1/2}$ only depend on $\xi_1, \ldots, \xi_{t-1}$. For any $\tau, \lambda > 0$, by Assumption 5.1 with $\mathbf{v} = \tau \cdot \alpha_{t-1} \widehat{\mathbf{V}}_{t-1}^{-1/2} \nabla f(\mathbf{x}_t)$, we have

$$\mathbb{E}\left\{ \exp\left[ \lambda \alpha_{t-1} \nabla f(\mathbf{x}_t)^\top \widehat{\mathbf{V}}_{t-1}^{-1/2} (\mathbf{g}_t - \nabla f(\mathbf{x}_t)) \right] \Big| \mathcal{F}_{t-1} \right\} \leq \exp(\sigma^2 \alpha_{t-1}^2 \lambda^2 \|\widehat{\mathbf{V}}_{t-1}^{-1/2} \nabla f(\mathbf{x}_t)\|_2^2 / 2).$$

Denote $Z_t = \alpha_{t-1} \nabla f(\mathbf{x}_t)^\top \widehat{\mathbf{V}}_{t-1}^{-1/2} (\mathbf{g}_t - \nabla f(\mathbf{x}_t))$. Then we have

$$\begin{aligned}
\mathbb{P}(Z_t \geq \tau | \mathcal{F}_{t-1}) &= \mathbb{P}[\exp(\lambda Z_t) \geq \exp(\lambda \tau) | \mathcal{F}_{t-1}] \\
&= \mathbb{E}[\mathbb{1}\{\exp(-\lambda \tau + \lambda Z_t) \geq 1\} | \mathcal{F}_{t-1}] \\
&\leq \exp(-\lambda \tau) \cdot \mathbb{E}[\exp(\lambda Z_t) | \mathcal{F}_{t-1}] \\
&\leq \exp(-\lambda \tau) \cdot \exp(\sigma^2 \alpha_{t-1}^2 \lambda^2 \|\widehat{\mathbf{V}}_{t-1}^{-1/2} \nabla f(\mathbf{x}_t)\|_2^2 / 2) \\
&= \exp(-\lambda \tau + \sigma^2 \alpha_{t-1}^2 \lambda^2 \|\widehat{\mathbf{V}}_{t-1}^{-1/2} \nabla f(\mathbf{x}_t)\|_2^2 / 2).
\end{aligned}$$

With exactly the same proof, we also have

$$\mathbb{P}(Z_t \leq -\tau | \mathcal{F}_{t-1}) \leq \exp(-\lambda \tau + \sigma^2 \alpha_{t-1}^2 \lambda^2 \|\widehat{\mathbf{V}}_{t-1}^{-1/2} \nabla f(\mathbf{x}_t)\|_2^2 / 2).$$

Combining above two inequalities, we have

$$\mathbb{P}(|Z_t| \geq \tau | \mathcal{F}_{t-1}) \leq 2 \exp(-\lambda \tau + \sigma^2 \alpha_{t-1}^2 \lambda^2 \|\widehat{\mathbf{V}}_{t-1}^{-1/2} \nabla f(\mathbf{x}_t)\|_2^2 / 2).$$

Choosing $\lambda = [\sigma^2 \alpha_{t-1}^2 \|\widehat{\mathbf{V}}_{t-1}^{-1/2} \nabla f(\mathbf{x}_t)\|_2^2]^{-1} \tau$, we finally obtain

$$\mathbb{P}(|Z_t| \geq \tau | \mathcal{F}_{t-1}) \leq 2 \exp(-\tau^2 / (2\sigma_t^2)) \tag{A.16}$$

for all $\tau > 0$, where $\sigma_t = \sigma \alpha_{t-1} \|\widehat{\mathbf{V}}_{t-1}^{-1/2} \nabla f(\mathbf{x}_t)\|_2$. The tail bound (A.16) enables the application of Lemma 6 in Jin et al. (2019), which gives that with probability at least $1 - \delta$,

$$\left| \sum_{t=2}^{T} Z_t \right| \leq \sum_{t=2}^{T} \sigma_t^2 + C \log(2/\epsilon),$$

where $C$ is an absolute constant. Plugging in the definitions of $Z_t$ and $\sigma_t$, we obtain

$$\begin{aligned}
&\left| \sum_{t=2}^{T} \alpha_{t-1} \nabla f(\mathbf{x}_t)^\top \widehat{\mathbf{V}}_{t-1}^{-1/2} \mathbf{g}_t - \sum_{t=2}^{T} \alpha_{t-1} \nabla f(\mathbf{x}_t)^\top \widehat{\mathbf{V}}_{t-1}^{-1/2} \nabla f(\mathbf{x}_t) \right| \\
&\leq \sum_{t=2}^{T} \sigma^2 \alpha_{t-1}^2 \|\widehat{\mathbf{V}}_{t-1}^{-1/2} \nabla f(\mathbf{x}_t)\|_2^2 + C \log(2/\delta) \\
&\leq \epsilon^{-1} \sigma^2 \sum_{t=2}^{T} \alpha_{t-1}^2 \|\nabla f(\mathbf{x}_t)\|_2^2 + C \log(2/\delta),
\end{aligned} \tag{A.17}$$

where the second inequality is by the fact that the diagonal entries of $\widehat{\mathbf{V}}_{t-1}$ are all loewr bounded by $\epsilon$. Substituting (A.14), (A.15) and (A.17) into (A.13), we have

$$\begin{aligned}
\sum_{t=2}^{T} \alpha_{t-1} \nabla f(\mathbf{x}_t)^\top \widehat{\mathbf{V}}_{t-1}^{-1/2} \nabla f(\mathbf{x}_t) &\leq \Delta + \frac{G_\infty^2 \alpha_1 \epsilon^{-1/2} d}{1 - \beta_1} + d\alpha_1 G_\infty + \frac{LT^{1/2} \alpha_t^2}{\epsilon^{1/2}(1 - \beta_2)^{1/2}(1 - \gamma)} \sum_{i=1}^{d} \|\mathbf{g}_{1:T,i}\|_2 \\
&\quad + \left( \frac{\beta_1}{1 - \beta_1} \right)^2 \frac{2LT^{1/2} \alpha_t^2 (1 - \beta_1)}{\epsilon^{1/2}(1 - \beta_2)^{1/2}(1 - \gamma)} \sum_{i=1}^{d} \|\mathbf{g}_{1:T,i}\|_2 \\
&\quad + \epsilon^{-1} \sigma^2 \sum_{t=2}^{T} \alpha_{t-1}^2 \|\nabla f(\mathbf{x}_t)\|_2^2 + C \log(2/\delta).
\end{aligned}$$

Moreover, by Lemma A.1, we have $\nabla f(\mathbf{x}_t)^\top \widehat{\mathbf{V}}_{t-1}^{-1/2} \nabla f(\mathbf{x}_t) \geq (G_\infty + \sqrt{\epsilon})^{-1} \|\nabla f(\mathbf{x}_t)\|_2^2$, and therefore by choosing $\alpha_t = \alpha$ and rearranging terms, we have

$$(G_\infty + \sqrt{\epsilon})^{-1} \sum_{t=2}^{T} \alpha(1 - \epsilon^{-1}\sigma^2\alpha)\|\nabla f(\mathbf{x}_t)\|_2^2$$

$$\leq \Delta + \frac{G_\infty^2 \alpha \epsilon^{-1/2} d}{1 - \beta_1} + d\alpha G_\infty + \frac{LT^{1/2}\alpha^2}{\epsilon^{1/2}(1-\beta_2)^{1/2}(1-\gamma)} \sum_{i=1}^{d} \|\mathbf{g}_{1:T,i}\|_2$$

$$+ \left(\frac{\beta_1}{1-\beta_1}\right)^2 \frac{2LT^{1/2}\alpha^2(1-\beta_1)}{\epsilon^{1/2}(1-\beta_2)^{1/2}(1-\gamma)} \sum_{i=1}^{d} \|\mathbf{g}_{1:T,i}\|_2 + C\log(2/\delta).$$

Therefore when $\alpha < \sigma^{-2}\epsilon/2$, we have

$$\frac{1}{T-1} \sum_{t=2}^{T} \|\nabla f(\mathbf{x}_t)\|_2^2$$

$$\leq \frac{4(G_\infty + \sqrt{\epsilon})}{T\alpha} \cdot \Delta + \frac{4G_\infty^2(G_\infty + \sqrt{\epsilon})\epsilon^{-1/2}}{1-\beta_1} \cdot \frac{d}{T} + 4G_\infty(G_\infty + \sqrt{\epsilon}) \cdot \frac{d}{T}$$

$$+ \frac{4(G_\infty + \sqrt{\epsilon})L\alpha}{\epsilon^{1/2}(1-\beta_2)^{1/2}(1-\gamma)T^{1/2}} \sum_{i=1}^{d} \|\mathbf{g}_{1:T,i}\|_2$$

$$+ \left(\frac{\beta_1}{1-\beta_1}\right)^2 \frac{8(G_\infty + \sqrt{\epsilon})L\alpha(1-\beta_1)}{\epsilon^{1/2}(1-\beta_2)^{1/2}(1-\gamma)T^{1/2}} \sum_{i=1}^{d} \|\mathbf{g}_{1:T,i}\|_2 + \frac{C'(G_\infty + \sqrt{\epsilon})\log(2/\delta)}{T\alpha},$$

where $C'$ is an absolute constant.

Now by the theorem condition $\|\mathbf{g}_{1:T,i}\|_2 \leq G_\infty T^s$, we have

$$\frac{1}{T-1} \sum_{t=2}^{T} \|\nabla f(\mathbf{x}_t)\|_2^2 \leq \frac{M_1}{T\alpha} + \frac{M_2 d}{T} + \frac{\alpha M_3 d}{T^{1/2-s}},$$

where $\{M_i\}_{i=1}^3$ are defined in Theorem 5.2. This completes the proof. ∎

## A.5 Proof of Corollary 5.4

**Proof** [Proof of Corollary 5.4] Following the proof for Theorem 5.2, setting $\beta_1' = \beta_1 = 0$ and $\beta_2' = \beta_2 = \beta$ in Lemma A.2 we get the conclusion. ∎

## A.6 Proof of Corollary 5.4

**Proof** [Proof of Corollary 5.4] Following the proof for Theorem 5.2, setting $\beta_1' = \beta_1 = 0$, $\beta_2 = 1$ and $\beta_2' = 0$ in Lemma A.2 we get the conclusion. ∎

# B Proof of Technical Lemmas

## B.1 Proof of Lemma A.1

**Proof** [Proof of Lemma A.1] Since $f$ has $G_\infty$-bounded stochastic gradient, for any $\mathbf{x}$ and $\xi$, $\|\nabla f(\mathbf{x}; \xi)\|_\infty \leq G_\infty$. Thus, we have

$$\|\nabla f(\mathbf{x})\|_\infty = \|\mathbb{E}_\xi \nabla f(\mathbf{x}; \xi)\|_\infty \leq \mathbb{E}_\xi \|\nabla f(\mathbf{x}; \xi)\|_\infty \leq G_\infty.$$

Next we bound $\|\mathbf{m}_t\|_\infty$. We have $\|\mathbf{m}_0\|_\infty = 0 \leq G_\infty$. Suppose that $\|\mathbf{m}_t\|_\infty \leq G_\infty$, then for $\mathbf{m}_{t+1}$, we have

$$\begin{aligned}
\|\mathbf{m}_{t+1}\|_\infty &= \|\beta_1 \mathbf{m}_t + (1 - \beta_1)\mathbf{g}_{t+1}\|_\infty \\
&\leq \beta_1 \|\mathbf{m}_t\|_\infty + (1 - \beta_1)\|\mathbf{g}_{t+1}\|_\infty \\
&\leq \beta_1 G_\infty + (1 - \beta_1)G_\infty \\
&= G_\infty.
\end{aligned}$$

Thus, for any $t \geq 0$, we have $\|\mathbf{m}_t\|_\infty \leq G_\infty$. Finally we bound $\|\widehat{\mathbf{v}}_t\|_\infty$. First we have $\|\mathbf{v}_0\|_\infty = \|\widehat{\mathbf{v}}_0\|_\infty = 0 \leq G_\infty^2$. Suppose that $\|\widehat{\mathbf{v}}_t\|_\infty \leq G_\infty^2$ and $\|\mathbf{v}_t\|_\infty \leq G_\infty^2$. Note that we have

$$\begin{aligned}
\|\mathbf{v}_{t+1}\|_\infty &= \|\beta_2 \mathbf{v}_t + (1 - \beta_2)\mathbf{g}_{t+1}^2\|_\infty \\
&\leq \beta_2 \|\mathbf{v}_t\|_\infty + (1 - \beta_2)\|\mathbf{g}_{t+1}^2\|_\infty \\
&\leq \beta_2 G_\infty^2 + (1 - \beta_2)G_\infty^2 \\
&= G_\infty^2,
\end{aligned}$$

and by definition, we have $\|\widehat{\mathbf{v}}_{t+1}\|_\infty = \max\{\|\widehat{\mathbf{v}}_t\|_\infty, \|\mathbf{v}_{t+1}\|_\infty\} \leq G_\infty^2$. Thus, for any $t \geq 0$, we have $\|\widehat{\mathbf{v}}_t\|_\infty \leq G_\infty^2$. ∎

## B.2 Proof of Lemma A.2

**Proof** Recall that $\widehat{v}_{t,j}, m_{t,j}, g_{t,j}$ denote the $j$-th coordinate of $\widehat{\mathbf{v}}_t, \mathbf{m}_t$ and $\mathbf{g}_t$. We have

$$\begin{aligned}
\alpha_t^2 \|\widehat{\mathbf{V}}_t^{-1/2} \mathbf{m}_t\|_2^2 &= \alpha_t^2 \sum_{i=1}^d \frac{m_{t,i}^2}{\widehat{v}_{t,i}^{1/2}} \cdot \frac{\widehat{v}_{t,i}^{1/2}}{\widehat{v}_{t,i} + \epsilon} \\
&\leq \alpha_t^2 \sum_{i=1}^d \frac{m_{t,i}^2}{\widehat{v}_{t,i}^{1/2}} \cdot \frac{\widehat{v}_{t,i}^{1/2}}{2\widehat{v}_{t,i}^{1/2} \epsilon^{1/2}} \\
&\leq \frac{\alpha_t^2}{2\epsilon^{1/2}} \sum_{i=1}^d \frac{m_{t,i}^2}{v_{t,i}^{1/2}} \\
&= \frac{\alpha_t^2}{2\epsilon^{1/2}} \sum_{i=1}^d \frac{(\sum_{j=1}^t (1 - \beta_1')\beta_1^{t-j} g_{j,i})^2}{(\sum_{j=1}^t (1 - \beta_2')\beta_2^{t-j} g_{j,i}^2)^{1/2}},
\end{aligned} \tag{B.1}$$

where the first inequality holds since $a + b \geq 2\sqrt{ab}$ and the second inequality holds because $\widehat{v}_{t,i} \geq v_{t,i}$. Next we have

$$\begin{aligned}
\frac{\alpha_t^2}{2\epsilon^{1/2}} \sum_{i=1}^d \frac{(\sum_{j=1}^t (1 - \beta_1')\beta_1^{t-j} g_{j,i})^2}{(\sum_{j=1}^t (1 - \beta_2')\beta_2^{t-j} g_{j,i}^2)^{1/2}} &\leq \frac{\alpha_t^2 (1 - \beta_1')^2}{2\epsilon^{1/2}(1 - \beta_2')^{1/2}} \sum_{i=1}^d \frac{(\sum_{j=1}^t \beta_1^{t-j})(\sum_{j=1}^t \beta_1^{t-j} |g_{j,i}|^2)}{(\sum_{j=1}^t \beta_2^{t-j} g_{j,i}^2)^{1/2}} \\
&\leq \frac{\alpha_t^2 (1 - \beta_1')}{2\epsilon^{1/2}(1 - \beta_2')^{1/2}} \sum_{i=1}^d \frac{\sum_{j=1}^t \beta_1^{t-j} |g_{j,i}|^2}{(\sum_{j=1}^t \beta_2^{t-j} g_{j,i}^2)^{1/2}},
\end{aligned} \tag{B.2}$$

where the first inequality holds due to Cauchy inequality, and the last inequality holds because $\sum_{j=1}^{t} \beta_1^{t-j} \leq (1-\beta_1)^{-1}$. Note that

$$\sum_{i=1}^{d} \frac{\sum_{j=1}^{t} \beta_1^{t-j}|g_{j,i}|^2}{(\sum_{j=1}^{t} \beta_2^{t-j} g_{j,i}^2)^{1/2}} \leq \sum_{i=1}^{d} \sum_{j=1}^{t} \frac{\beta_1^{t-j}|g_{j,i}|^2}{(\beta_2^{t-j} g_{j,i}^2)^{1/2}}$$

$$= \sum_{i=1}^{d} \sum_{j=1}^{t} \gamma^{t-j}|g_{j,i}|, \tag{B.3}$$

where the equality holds due to the definition of $\gamma$. Substituting (B.2) and (B.3) into (B.1), we have

$$\alpha_t^2 \|\widehat{\mathbf{V}}_t^{-1/2} \mathbf{m}_t\|_2^2 \leq \frac{\alpha_t^2(1-\beta_1')}{2\epsilon^{1/2}(1-\beta_2')^{1/2}} \sum_{i=1}^{d} \sum_{j=1}^{t} \gamma^{t-j}|g_{j,i}|. \tag{B.4}$$

Telescoping (B.4) for $t = 1$ to $T$, we have

$$\sum_{t=1}^{T} \alpha_t^2 \|\widehat{\mathbf{V}}_t^{-1/2} \mathbf{m}_t\|_2^2 \leq \frac{\alpha_t^2(1-\beta_1')}{2\epsilon^{1/2}(1-\beta_2')^{1/2}} \sum_{t=1}^{T} \sum_{i=1}^{d} \sum_{j=1}^{t} \gamma^{t-j}|g_{j,i}|$$

$$= \frac{\alpha_t^2(1-\beta_1')}{2\epsilon^{1/2}(1-\beta_2')^{1/2}} \sum_{i=1}^{d} \sum_{j=1}^{T} |g_{j,i}| \sum_{t=j}^{T} \gamma^{t-j}$$

$$\leq \frac{\alpha_t^2(1-\beta_1')}{2\epsilon^{1/2}(1-\beta_2')^{1/2}(1-\gamma)} \sum_{i=1}^{d} \sum_{j=1}^{T} |g_{j,i}|. \tag{B.5}$$

Finally, we have

$$\sum_{i=1}^{d} \sum_{j=1}^{T} |g_{j,i}| \leq \sum_{i=1}^{d} \left( \sum_{j=1}^{T} g_{j,i}^2 \right)^{1/2} \cdot T^{1/2} = T^{1/2} \sum_{i=1}^{d} \|\mathbf{g}_{1:T,i}\|_2, \tag{B.6}$$

where the inequality holds due to Hölder's inequality. Substituting (B.6) into (B.5), we have

$$\sum_{t=1}^{T} \alpha_t^2 \|\widehat{\mathbf{V}}_t^{-1/2} \mathbf{m}_t\|_2^2 \leq \frac{T^{1/2}\alpha_t^2(1-\beta_1')}{2\epsilon^{1/2}(1-\beta_2')^{1/2}(1-\gamma)} \sum_{i=1}^{d} \|\mathbf{g}_{1:T,i}\|_2.$$

Specifically, taking $\beta_1 = 0$, we have $\mathbf{m}_t = \mathbf{g}_t$, then

$$\sum_{t=1}^{T} \alpha_t^2 \|\widehat{\mathbf{V}}_t^{-1/2} \mathbf{g}_t\|_2^2 \leq \frac{T^{1/2}\alpha_t^2}{2\epsilon^{1/2}(1-\beta_2')^{1/2}(1-\gamma)} \sum_{i=1}^{d} \|\mathbf{g}_{1:T,i}\|_2.$$

$\blacksquare$

### B.3 Proof of Lemma A.3

**Proof** By definition, we have

$$\mathbf{z}_{t+1} = \mathbf{x}_{t+1} + \frac{\beta_1}{1-\beta_1}(\mathbf{x}_{t+1} - \mathbf{x}_t)$$

$$= \frac{1}{1-\beta_1}\mathbf{x}_{t+1} - \frac{\beta_1}{1-\beta_1}\mathbf{x}_t.$$

Then we have

$$\mathbf{z}_{t+1} - \mathbf{z}_t = \frac{1}{1-\beta_1}(\mathbf{x}_{t+1} - \mathbf{x}_t) - \frac{\beta_1}{1-\beta_1}(\mathbf{x}_t - \mathbf{x}_{t-1})$$
$$= \frac{1}{1-\beta_1}\big(-\alpha_t\widehat{\mathbf{V}}_t^{-1/2}\mathbf{m}_t\big) + \frac{\beta_1}{1-\beta_1}\alpha_{t-1}\widehat{\mathbf{V}}_{t-1}^{-1/2}\mathbf{m}_{t-1}.$$

The equities above are based on definition. Then we have

$$\mathbf{z}_{t+1} - \mathbf{z}_t = \frac{-\alpha_t\widehat{\mathbf{V}}_t^{-1/2}}{1-\beta_1}\Big[\beta_1\mathbf{m}_{t-1} + (1-\beta_1)\mathbf{g}_t\Big] + \frac{\beta_1}{1-\beta_1}\alpha_{t-1}\widehat{\mathbf{V}}_{t-1}^{-1/2}\mathbf{m}_{t-1}$$
$$= \frac{\beta_1}{1-\beta_1}\big(\alpha_{t-1}\widehat{\mathbf{V}}_{t-1}^{-1/2} - \alpha_t\widehat{\mathbf{V}}_t^{-1/2}\big)\mathbf{m}_{t-1} - \alpha_t\widehat{\mathbf{V}}_t^{-1/2}\mathbf{g}_t$$
$$= \frac{\beta_1}{1-\beta_1}\alpha_{t-1}\widehat{\mathbf{V}}_{t-1}^{-1/2}\Big[\mathbf{I} - \big(\alpha_t\widehat{\mathbf{V}}_t^{-1/2}\big)\big(\alpha_{t-1}\widehat{\mathbf{V}}_{t-1}^{-1/2}\big)^{-1}\Big]\mathbf{m}_{t-1} - \alpha_t\widehat{\mathbf{V}}_t^{-1/2}\mathbf{g}_t$$
$$= \frac{\beta_1}{1-\beta_1}\Big[\mathbf{I} - \big(\alpha_t\widehat{\mathbf{V}}_t^{-1/2}\big)\big(\alpha_{t-1}\widehat{\mathbf{V}}_{t-1}^{-1/2}\big)^{-1}\Big](\mathbf{x}_{t-1} - \mathbf{x}_t) - \alpha_t\widehat{\mathbf{V}}_t^{-1/2}\mathbf{g}_t.$$

The equalities above follow by combining the like terms. ∎

### B.4 Proof of Lemma A.4

**Proof** By Lemma A.3, we have

$$\|\mathbf{z}_{t+1} - \mathbf{z}_t\|_2 = \left\|\frac{\beta_1}{1-\beta_1}\Big[\mathbf{I} - (\alpha_t\widehat{\mathbf{V}}_t^{-1/2})(\alpha_{t-1}\widehat{\mathbf{V}}_{t-1}^{-1/2})^{-1}\Big](\mathbf{x}_{t-1} - \mathbf{x}_t) - \alpha_t\widehat{\mathbf{V}}_t^{-1/2}\mathbf{g}_t\right\|_2$$
$$\le \frac{\beta_1}{1-\beta_1}\left\|\mathbf{I} - (\alpha_t\widehat{\mathbf{V}}_t^{-1/2})(\alpha_{t-1}\widehat{\mathbf{V}}_{t-1}^{-1/2})^{-1}\right\|_{\infty,\infty}\cdot\|\mathbf{x}_{t-1} - \mathbf{x}_t\|_2 + \left\|\alpha\widehat{\mathbf{V}}_t^{-1/2}\mathbf{g}_t\right\|_2,$$

where the inequality holds because the term $\beta_1/(1-\beta_1)$ is positive, and triangle inequality. Considering that $\alpha_t\widehat{\mathbf{v}}_{t,j}^{-1/2} \le \alpha_{t-1}\widehat{\mathbf{v}}_{t-1,j}^{-1/2}$, when $p > 0$, we have $\left\|\mathbf{I} - (\alpha_t\widehat{\mathbf{V}}_t^{-1/2})(\alpha_{t-1}\widehat{\mathbf{V}}_{t-1}^{-1/2})^{-1}\right\|_{\infty,\infty} \le 1$. With that fact, the term above can be bound as:

$$\|\mathbf{z}_{t+1} - \mathbf{z}_t\|_2 \le \left\|\alpha\widehat{\mathbf{V}}_t^{-1/2}\mathbf{g}_t\right\|_2 + \frac{\beta_1}{1-\beta_1}\|\mathbf{x}_{t-1} - \mathbf{x}_t\|_2.$$

This completes the proof. ∎

### B.5 Proof of Lemma A.5

**Proof** For term $\|\nabla f(\mathbf{z}_t) - \nabla f(\mathbf{x}_t)\|_2$, we have:

$$\|\nabla f(\mathbf{z}_t) - \nabla f(\mathbf{x}_t)\|_2 \le L\|\mathbf{z}_t - \mathbf{x}_t\|_2 \le L\left\|\frac{\beta_1}{1-\beta_1}(\mathbf{x}_t - \mathbf{x}_{t-1})\right\|_2 \le L\Big(\frac{\beta_1}{1-\beta_1}\Big)\cdot\|\mathbf{x}_t - \mathbf{x}_{t-1}\|_2,$$

where the last inequality holds because the term $\beta_1/(1-\beta_1)$ is positive. ∎

## B.6 Proof of Lemma A.6

**Proof** Since $f$ is $L$-smooth, we have:

$$
\begin{aligned}
f(\mathbf{z}_{t+1}) &\leq f(\mathbf{z}_t) + \nabla f(\mathbf{z}_t)^\top (\mathbf{z}_{t+1} - \mathbf{z}_t) + \frac{L}{2}\|\mathbf{z}_{t+1} - \mathbf{z}_t\|_2^2 \\
&= f(\mathbf{z}_t) + \underbrace{\nabla f(\mathbf{x}_t)^\top (\mathbf{z}_{t+1} - \mathbf{z}_t)}_{I_1} + \underbrace{(\nabla f(\mathbf{z}_t) - \nabla f(\mathbf{x}_t))^\top (\mathbf{z}_{t+1} - \mathbf{z}_t)}_{I_2} + \underbrace{\frac{L}{2}\|\mathbf{z}_{t+1} - \mathbf{z}_t\|_2^2}_{I_3}
\end{aligned}
\tag{B.7}
$$

In the following, we bound $I_1$, $I_2$ and $I_3$ separately.

**Bounding term $I_1$:** When $t = 1$, we have

$$
\nabla f(\mathbf{x}_1)^\top (\mathbf{z}_2 - \mathbf{z}_1) = -\nabla f(\mathbf{x}_1)^\top \alpha_1 \widehat{\mathbf{V}}_t^{-1/2} \mathbf{g}_1.
\tag{B.8}
$$

For $t \geq 2$, we have

$$
\begin{aligned}
\nabla f(\mathbf{x}_t)^\top (\mathbf{z}_{t+1} - \mathbf{z}_t) &= \nabla f(\mathbf{x}_t)^\top \left[ \frac{\beta_1}{1-\beta_1} \big( \alpha_{t-1}\widehat{\mathbf{V}}_{t-1}^{-1/2} - \alpha_t\widehat{\mathbf{V}}_t^{-1/2} \big) \mathbf{m}_{t-1} - \alpha_t \widehat{\mathbf{V}}_t^{-1/2} \mathbf{g}_t \right] \\
&= \frac{\beta_1}{1-\beta_1} \nabla f(\mathbf{x}_t)^\top \big( \alpha_{t-1}\widehat{\mathbf{V}}_{t-1}^{-1/2} - \alpha_t\widehat{\mathbf{V}}_t^{-1/2} \big) \mathbf{m}_{t-1} - \nabla f(\mathbf{x}_t)^\top \alpha_t \widehat{\mathbf{V}}_t^{-1/2} \mathbf{g}_t,
\end{aligned}
\tag{B.9}
$$

where the first equality holds due to (A.3) in Lemma A.3. For $\nabla f(\mathbf{x}_t)^\top (\alpha_{t-1}\widehat{\mathbf{V}}_{t-1}^{-1/2} - \alpha_t\widehat{\mathbf{V}}_t^{-1/2})\mathbf{m}_{t-1}$ in (B.9), we have

$$
\begin{aligned}
\nabla f(\mathbf{x}_t)^\top (\alpha_{t-1}\widehat{\mathbf{V}}_{t-1}^{-1/2} - \alpha_t\widehat{\mathbf{V}}_t^{-1/2})\mathbf{m}_{t-1} &\leq \|\nabla f(\mathbf{x}_t)\|_\infty \cdot \big\|\alpha_{t-1}\widehat{\mathbf{V}}_{t-1}^{-1/2} - \alpha_t\widehat{\mathbf{V}}_t^{-1/2}\big\|_{1,1} \cdot \|\mathbf{m}_{t-1}\|_\infty \\
&\leq G_\infty^2 \Big[ \big\|\alpha_{t-1}\widehat{\mathbf{V}}_{t-1}^{-1/2}\big\|_{1,1} - \big\|\alpha_t\widehat{\mathbf{V}}_t^{-1/2}\big\|_{1,1} \Big].
\end{aligned}
\tag{B.10}
$$

The first inequality holds because for a positive diagonal matrix $\mathbf{A}$, we have $\mathbf{x}^\top \mathbf{A}\mathbf{y} \leq \|\mathbf{x}\|_\infty \cdot \|\mathbf{A}\|_{1,1} \cdot \|\mathbf{y}\|_\infty$. The second inequality holds due to $\alpha_{t-1}\widehat{\mathbf{V}}_{t-1}^{-1/2} \succeq \alpha_t\widehat{\mathbf{V}}_t^{-1/2} \succeq 0$. Next we bound $-\nabla f(\mathbf{x}_t)^\top \alpha_t\widehat{\mathbf{V}}_t^{-1/2}\mathbf{g}_t$. We have

$$
\begin{aligned}
-\nabla f(\mathbf{x}_t)^\top \alpha_t\widehat{\mathbf{V}}_t^{-1/2}\mathbf{g}_t &= -\nabla f(\mathbf{x}_t)^\top \alpha_{t-1}\widehat{\mathbf{V}}_{t-1}^{-1/2}\mathbf{g}_t - \nabla f(\mathbf{x}_t)^\top \big( \alpha_t\widehat{\mathbf{V}}_t^{-1/2} - \alpha_{t-1}\widehat{\mathbf{V}}_{t-1}^{-1/2} \big)\mathbf{g}_t \\
&\leq -\nabla f(\mathbf{x}_t)^\top \alpha_{t-1}\widehat{\mathbf{V}}_{t-1}^{-1/2}\mathbf{g}_t + \|\nabla f(\mathbf{x}_t)\|_\infty \cdot \big\|\alpha_t\widehat{\mathbf{V}}_t^{-1/2} - \alpha_{t-1}\widehat{\mathbf{V}}_{t-1}^{-1/2}\big\|_{1,1} \cdot \|\mathbf{g}_t\|_\infty \\
&\leq -\nabla f(\mathbf{x}_t)^\top \alpha_{t-1}\widehat{\mathbf{V}}_{t-1}^{-1/2}\mathbf{g}_t + G_\infty^2 \big( \big\|\alpha_{t-1}\widehat{\mathbf{V}}_{t-1}^{-1/2}\big\|_{1,1} - \big\|\alpha_t\widehat{\mathbf{V}}_t^{-1/2}\big\|_{1,1} \big).
\end{aligned}
\tag{B.11}
$$

The first inequality holds because for a positive diagonal matrix $\mathbf{A}$, we have $\mathbf{x}^\top \mathbf{A}\mathbf{y} \leq \|\mathbf{x}\|_\infty \cdot \|\mathbf{A}\|_{1,1} \cdot \|\mathbf{y}\|_\infty$. The second inequality holds due to $\alpha_{t-1}\widehat{\mathbf{V}}_{t-1}^{-1/2} \succeq \alpha_t\widehat{\mathbf{V}}_t^{-1/2} \succeq 0$. Substituting (B.10) and (B.11) into (B.9), we have

$$
\nabla f(\mathbf{x}_t)^\top (\mathbf{z}_{t+1} - \mathbf{z}_t) \leq -\nabla f(\mathbf{x}_t)^\top \alpha_{t-1}\widehat{\mathbf{V}}_{t-1}^{-1/2}\mathbf{g}_t + \frac{1}{1-\beta_1} G_\infty^2 \big( \big\|\alpha_{t-1}\widehat{\mathbf{V}}_{t-1}^{-1/2}\big\|_{1,1} - \big\|\alpha_t\widehat{\mathbf{V}}_t^{-1/2}\big\|_{1,1} \big).
\tag{B.12}
$$

**Bounding term $I_2$:** For $t \geq 1$, we have

$$
\begin{aligned}
\big(\nabla f(\mathbf{z}_t) - \nabla f(\mathbf{x}_t)\big)^\top (\mathbf{z}_{t+1} - \mathbf{z}_t) &\leq \big\|\nabla f(\mathbf{z}_t) - \nabla f(\mathbf{x}_t)\big\|_2 \cdot \|\mathbf{z}_{t+1} - \mathbf{z}_t\|_2 \\
&\leq \Big( \big\|\alpha_t\widehat{\mathbf{V}}_t^{-1/2}\mathbf{g}_t\big\|_2 + \frac{\beta_1}{1-\beta_1}\|\mathbf{x}_{t-1} - \mathbf{x}_t\|_2 \Big) \cdot \frac{\beta_1}{1-\beta_1} \cdot L\|\mathbf{x}_t - \mathbf{x}_{t-1}\|_2 \\
&= L\frac{\beta_1}{1-\beta_1}\big\|\alpha_t\widehat{\mathbf{V}}_t^{-1/2}\mathbf{g}_t\big\|_2 \cdot \|\mathbf{x}_t - \mathbf{x}_{t-1}\|_2 + L\Big(\frac{\beta_1}{1-\beta_1}\Big)^2 \|\mathbf{x}_t - \mathbf{x}_{t-1}\|_2^2 \\
&\leq L\big\|\alpha_t\widehat{\mathbf{V}}_t^{-1/2}\mathbf{g}_t\big\|_2^2 + 2L\Big(\frac{\beta_1}{1-\beta_1}\Big)^2 \|\mathbf{x}_t - \mathbf{x}_{t-1}\|_2^2,
\end{aligned}
\tag{B.13}
$$

where the second inequality holds because of Lemma A.3 and Lemma A.4, the last inequality holds due to Young's inequality.

**Bounding term $I_3$:** For $t \geq 1$, we have

$$\frac{L}{2}\|\mathbf{z}_{t+1} - \mathbf{z}_t\|_2^2 \leq \frac{L}{2}\Big[\big\|\alpha_t\widehat{\mathbf{V}}_t^{-1/2}\mathbf{g}_t\big\|_2 + \frac{\beta_1}{1-\beta_1}\|\mathbf{x}_{t-1} - \mathbf{x}_t\|_2\Big]^2$$

$$\leq L\big\|\alpha_t\widehat{\mathbf{V}}_t^{-1/2}\mathbf{g}_t\big\|_2^2 + 2L\Big(\frac{\beta_1}{1-\beta_1}\Big)^2\|\mathbf{x}_{t-1} - \mathbf{x}_t\|_2^2. \tag{B.14}$$

The first inequality is obtained by introducing Lemma A.3.

For $t = 1$, substituting (B.8), (B.13) and (B.14) into (B.7), taking expectation and rearranging terms, we have

$$f(\mathbf{z}_2) - f(\mathbf{z}_1) \leq -\nabla f(\mathbf{x}_1)^\top \alpha_1\widehat{\mathbf{V}}_1^{-1/2}\mathbf{g}_1 + 2L\big\|\alpha_1\widehat{\mathbf{V}}_1^{-1/2}\mathbf{g}_1\big\|_2^2 + 4L\Big(\frac{\beta_1}{1-\beta_1}\Big)^2\|\mathbf{x}_1 - \mathbf{x}_0\|_2^2$$

$$= -\nabla f(\mathbf{x}_1)^\top \alpha_1\widehat{\mathbf{V}}_1^{-1/2}\mathbf{g}_1 + 2L\big\|\alpha_1\widehat{\mathbf{V}}_1^{-1/2}\mathbf{g}_1\big\|_2^2$$

$$\leq d\alpha_1 G_\infty + 2L\big\|\alpha_1\widehat{\mathbf{V}}_1^{-1/2}\mathbf{g}_1\big\|_2^2, \tag{B.15}$$

where the last inequality holds because

$$-\nabla f(\mathbf{x}_1)^\top\widehat{\mathbf{V}}_1^{-1/2}\mathbf{g}_1 \leq d \cdot \|\nabla f(\mathbf{x}_1)\|_\infty \cdot \|\widehat{\mathbf{V}}_1^{-1/2}\mathbf{g}_1\|_\infty \leq dG_\infty.$$

For $t \geq 2$, substituting (B.12), (B.13) and (B.14) into (B.7), taking expectation and rearranging terms, we have

$$f(\mathbf{z}_{t+1}) + \frac{G_\infty^2\big\|\alpha_t\widehat{\mathbf{V}}_t^{-1/2}\big\|_{1,1}}{1-\beta_1} - \Big(f(\mathbf{z}_t) + \frac{G_\infty^2\big\|\alpha_{t-1}\widehat{\mathbf{V}}_{t-1}^{-1/2}\big\|_{1,1}}{1-\beta_1}\Big)$$

$$\leq -\nabla f(\mathbf{x}_t)^\top \alpha_{t-1}\widehat{\mathbf{V}}_{t-1}^{-1/2}\mathbf{g}_t + 2L\big\|\alpha_t\widehat{\mathbf{V}}_t^{-1/2}\mathbf{g}_t\big\|_2^2 + 4L\Big(\frac{\beta_1}{1-\beta_1}\Big)^2\|\mathbf{x}_t - \mathbf{x}_{t-1}\|_2^2,$$

which ends our proof. ∎

### B.7 Experimental Verification of the Growth Rate Condition

In order to show that the growth rate condition of the cumulative stochastic gradient indeed holds, we have conducted experiments to estimate the growth rate parameter $s$ for ResNet-18 (He et al., 2016) model and 3-layer LSTM model (Hochreiter & Schmidhuber, 1997) respectively. For simplicity, we assume $G_\infty = 1$ and estimate the growth rate $s$ by calculating the logarithm of the cumulative gradient norm $\log\|\mathbf{g}_{1:T,i}\|_2$ and calculate $\log\|\mathbf{g}_{1:T,i}\|_2$. As can be seen from Table 2, $s$ of adaptive gradient methods (AdaGrad, RMSProp and AMSGrad) is smaller than that of SGDM for training 3-layer LSTM model on the PennTreeBank (Marcus et al., 1993) dataset. All of them are actually far below the theoretical limit $1/2$ in this real experiment.

| method | $s$ | training loss | test perplexity |
|---|---|---|---|
| SGDM | 0.136 | 4.01 | 65.11 |
| AdaGrad | 0.089 | 3.92 | 64.90 |
| RMSProp | 0.085 | 3.84 | 63.77 |
| AMSGrad | 0.086 | 3.85 | 63.97 |

Table 2: Empirical growth rate parameter $s$ of 3-layer LSTM model on PennTreeBank dataset.

