# OpenReview forum: "On the Convergence of Adaptive Gradient Methods for Nonconvex Optimization"
_TMLR — Accepted by TMLR_

### Review · Reviewer_gkXY · 2023-12-26

**Summary Of Contributions:**

The paper presents significant contributions in nonconvex optimization, providing a detailed convergence analysis for a broad class of adaptive gradient methods, including AMSGrad, RMSProp, and AdaGrad. The authors propose new convergence guarantees both in expectation and high probability, which outperform the existing ones for adaptive gradient methods in terms of dimension. Indeed, Table 1 on page 3 clearly shows that the new convergence rates have a better dependence on the dimension $d$ and $T$ than the existing rates. It is also worth to note that the high probability bounds on the convergence rates established in this article are novel in nonconvex optimization and hold significant value, particularly when characterizing the performance of algorithms in a single run.

**Audience:**

Yes

**Broader Impact Concerns:**

There are no concerns on the ethical implications of the work.

**Claims And Evidence:**

Yes

**Requested Changes:**

There are some minor aspects which can be changed (they are not critical to securing recommendation for acceptance):


1. The reference to footnote number 3 in Table 1 in the text should be checked to ensure it is properly linked and functioning.

2. The missing dot at the end of the penultimate sentence on page 14 can be added to maintain proper punctuation.

3. In section B.6, additional details about estimating the growth rate parameter $s$ could be included to provide a more comprehensive explanation of the estimation process.

**Strengths And Weaknesses:**

The strengths of this work are as follows:


1. The paper proposes a rigorous convergence analysis of adaptive gradient methods, including AMSGrad, RMSProp, and AdaGrad, supported by relevant evidence.

2. The paper establishes that for smooth nonconvex functions the analyzed adaptive gradient methods achieve better convergence rates in terms of dimension when compared to the existing results for adaptive gradient methods.

3. By proving high probability bounds on the convergence rates, in addition to bounds in expectation, the work introduces novel and previously unestablished bounds.

---

> ### Author Response · Authors · 2024-01-23
> **To Reviewer gkXY**
>
> Thank you for your helpful feedback!
>
> **Q1:** The reference to footnote number 3 in Table 1 in the text should be checked to ensure it is properly linked and functioning.
>
> **A1:** Thanks for pointing out. We have fixed them in revision.
>
> **Q2:** The missing dot at the end of the penultimate sentence on page 14 can be added to maintain proper punctuation.
>
> **A2:** Thanks for pointing out. We have fixed them in revision.
>
> **Q3:** In section B.6, additional details about estimating the growth rate parameter $s$ could be included to provide a more comprehensive explanation of the estimation process.
>
> **A3:** Thanks for the question and we are happy to provide more details. Specifically, the growth rate condition states that $ \|g_{1:t,i}\|\_2 \leq G_{\infty} T^s $. For simplicity, we set $G_{\infty} = 1$ as our observation didn’t find any single gradient entry from all iterations that goes beyond 1. Then we can estimate $s$ by directly solving the equation  $\|g_{1:T,i}\|_2 = T^s$ for each dimension. The final estimate of $s$ is selected as the averaged result over all dimensions. We have added it to Section B.6 in the revision.

---

### Review · Reviewer_6XWB · 2024-01-03

**Summary Of Contributions:**

The submission provides convergence rate analyses for adaptive gradient methods in nonconvex smooth stochastic optimization; more precisely, rates for the expected squared norm of the gradient for an iterate chosen at random. Both the assumptions and the rates are favorable, compared to existing works. Moreover, the work includes guarantees both in-expectation and with high-probability.

**Audience:**

Yes

**Claims And Evidence:**

Yes

**Requested Changes:**

I consider the following comments as essential in order to accept this submission.

Major comments:

1. Most discussions about the role of gradient sparsity ended up not making an impact on the results of the paper, e.g. in the form of a precise rate of convergence. I understand these discussions have already appeared in the seminal works on adaptive gradient methods, but I was left with the impression that these discussions were somewhat incomplete.
2. In all algorithms, it is not specified the range of the offset parameter $\epsilon$. Just looking at any of the upper bounds, it seems to always be beneficial to take $\epsilon\to\infty$ (or sufficiently large, to make the terms where $\epsilon^{-1/2}$ appears negligible. This seems intuitively wrong. More precisely, my questions are: Does $\epsilon$ have any hard constraints (other than nonnegativity)? If not, Why not making $\epsilon\to \infty$?
A slightly related question is the fact that some adaptive gradient methods need some offset parameter (see e.g., Figure 1 in the original adagrad paper https://www.jmlr.org/papers/volume12/duchi11a/duchi11a.pdf, or Section 3.5 in McMahan's survey https://www.jmlr.org/papers/volume18/14-428/14-428.pdf). Is the offset parameter in the studied results strictly necessary?

Minor comments:

1. Remark 4.4. I couldn't find anywhere in Ghadimi-Lan 2016 an upper bound matching (or resembling) the one in the manuscript. So the claim of "matching the rates of nonconvex SGD" seems unjustified. In the same remark, the rate is wrong: it should be $O(d^{1/2}/T^{3/4-s/2})$.
2. Notational inconsistency. The initial optimality gap is denoted as $\Delta$, but in all proofs the notation $\Delta f$ is used.
3. Equation (6.8) In the expression $(G_{\infty}^{2p})^{-1}$, where does $p$ come from?
4. Middle of page 20. There is a jump of line in the middle of the equation $\mathbb{P}[\exp(\lambda Z_t)\geq \exp(\lambda \tau)|{\cal F}_{t-1}]$.
5. Middle of page 24. $m_t(\alpha_{t-1}\hat V_{t-1}^{-1/2})$ is not possible as a matrix-vector product. Please switch the two terms. This error is present in multiple equations, so a more thorough revision within the proofs is needed.
6. Page 24 (and elsewhere). What is $\|\cdot\|_{\infty,\infty}$ and $\|\cdot\|_{1,1}$?

**Strengths And Weaknesses:**

Strengths:
1. Clear improvements upon existing work.
2. A unified treatment of different adaptive gradient methods.

Weaknesses:
1. One of the main advantages of adaptive gradient methods is avoiding the expensive hyperparameter tuning to optimize accuracy as a function of the stepsize. However, all results in this paper still contain hyperparameters (most crucially, $\alpha$), which needs to be set in a fairly nontrivial way to obtain the claimed rates.
2. While reading the manuscript, I missed a more high-level explanation of how these improvements were obtained. Namely, what major insights were obtained in this work that allowed the authors to get the improved dimension dependence.
3. Despite the paper providing a unifying framework for adaptive gradient methods, there is some amount of repetition in the arguments (see, e.g. the proofs of Thm 4.3 and Thm. 5.2). Ideally, these similar analyses could have been unified in a streamlined fashion.
4. Various typos and unclear notation make reading more difficult than it should be.

---

> ### Author Response · Authors · 2024-01-23
> **To Reviewer 6XWB**
>
> Thank you for your constructive comments.
>
> **Q1:** One of the main advantages of adaptive gradient methods is avoiding the expensive hyperparameter tuning to optimize accuracy as a function of the stepsize. However, all results in this paper still contain hyperparameters (most crucially, $\alpha$, which needs to be set in a fairly nontrivial way to obtain the claimed rates.
>
> **A1:** First, this is a paper focusing on the theoretical analysis of adaptive gradient methods and essentially all theoretical convergence analysis needs to have certain conditions on the learning rate for the desired rate. Being adaptive does not mean that they can work under any learning rate choice so it is perfectly normal to have a learning rate condition in the analysis. Second, in practice, the learning rate choice is always based on the hyperparameter tuning result rather than the theoretical values. In fact, the theoretical learning rate condition usually hides many problem-dependent and absolute constants (such as the smoothness parameter) that are impossible to know in practice. Therefore, we cannot calculate the exact learning rate condition and reason whether this requirement is too high or not.
>
>
> **Q2:** While reading the manuscript, I missed a more high-level explanation of how these improvements were obtained. Namely, what major insights were obtained in this work that allowed the authors to get the improved dimension dependence.
>
> **A2:** Intuitively speaking, the improvement of the convergence rate is due to an improved analysis that bounds the norm of momentums $\|\hat V_t^{-1/2}m_t\|$ and $\|\hat V_t^{-1/2}g_t\|$ by the norm of the cumulative stochastic gradient $|g_{1:T,i}|$, as we have shown in Lemma 6.5.
>
> **Q3:** Similar analyses could have been unified in a streamlined fashion.
>
> **A3:** Thanks for the suggestion. We have reorganized the structure of our proof to make things clear.
>
> **Q4:** Various typos and unclear notation make reading more difficult than it should be.
>
> **A4:** Thank you for pointing them out. We have fixed them.
>
> **Q5:** Most discussions about the role of gradient sparsity ended up not making an impact on the results of the paper, e.g. in the form of a precise rate of convergence.
>
> **A5:** We respectfully disagree with your comment ‘gradient sparsity not making an impact on the results of the paper’. In fact, our Remark 4.4 indeed showed that by carefully selecting the step size $\alpha$, it is possible to obtain a faster rate $1/T^{¾-s/2}$ compared with the vanilla rate $1/T^{1/2}$.
>
> **Q6:** Does $\epsilon$ have any hard constraints (other than nonnegativity)?
>
> **A6:** Thanks for pointing out this issue. In practice, $\epsilon$ is usually a small number for numerical stability purposes so it is often ignored in the analysis. Yet we can also show that $\epsilon$ cannot be chosen as infinity in the theoretical analysis. In fact, the spectral norm of the covariance matrix $|\hat V_{t-1}^{1/2}|$ is upper bounded by $G_\infty + \epsilon$ since $\hat V_{t-1}^{1/2}$ is a diagonal matrix whose diagonal entry is less than $G_\infty+\epsilon$, where $G_\infty$ comes from the gradient $g_t$ and $\epsilon$ comes from the offset parameter. This finally brings additional $\epsilon$ terms to the final convergence rate. The new convergence rate includes both $\epsilon$ and $\epsilon^{-1}$, therefore we can not set $\epsilon\rightarrow \infty$.
>
>
>
> **Q7:** A slightly related question is the fact that some adaptive gradient methods need some offset parameter. Is the offset parameter in the studied results strictly necessary?
>
> **A7:** The offset parameter is used to guarantee the covariance matrix $\hat V_t$ strictly positive to avoid the singular cases, therefore we believe it is necessary at least at the beginning stage of the adaptive optimization methods.
>
> **Q8:** Remark 4.4. I couldn't find anywhere in Ghadimi-Lan 2016 an upper bound matching (or resembling) the one in the manuscript. Meanwhile, there is a typo in the remark.
>
> **A8:** Corollary 3.a in Ghadimi and Lan, 2016 shows an upper bound $O(1/N + 1/\sqrt{N})$ to find a stationary point. We have fixed the typo in the revision.
>
> **Q9:** Equation (6.8) In the expression, where does $p$ come from?
>
> **A9:** Sorry for the confusion. This is a typo and we have fixed it in revision.
>
> **Q10:** Page 24 (and elsewhere). What is $|\cdot|\_{\infty,\infty}$ and $|\cdot|_{1,1}$?
>
>
>
> **A10:** We are sorry for the confusion. We have defined it in the Notation section (page 3). $|\cdot|\_{\infty, \infty}$ represents the maximized absolute value of the entries of the matrix and $|\cdot|_{1, 1}$ represents the summation of the absolute value of entries of the matrix.

---

> > ### Comment · Reviewer_6XWB · 2024-01-24
> > **Answer to authors' rebuttal**
> >
> > A1. While it is true that a fully adaptive method may be unattainable, e.g. Adagrad requires a single hyperparameter, namely the starting distance to the optimal solution. If I understand correctly, your results involve a number of other unknown parameters, and hence I can only expect the shape of this hyperparameter optimization problem to be substantially more complex (despite the fact that in both cases this optimization problem depends on a sine parameter, the shape of this function can become more complex due to the presence of other hidden parameters). Perhaps numerical experiments can support the benefits of your analysis, but currently I don't see the adaptive nature of the results.
> >
> > A2. Is it possible that you elaborate further on this in the paper?
> >
> > A3. Please add some explanatory details. By a quick look at the file I couldn't find these changes.
> >
> > A5. Thank you for the clarification.
> >
> > A6. This makes sense. Thank you.
> >
> > A8. Then I don't see how your rate matches the one in Ghadimi-Lan. That paper does not have any dimension dependence, whereas yours does.
> >
> > A10. Thanks for the clarification.

---

> ### Author Response · Authors · 2024-01-26
> **To Reviewer 6XWB's followup questions**
>
> Thank you for your followup questions.
>
> **Q1:** While it is true that a fully adaptive method may be unattainable, e.g. Adagrad requires a single hyperparameter, namely the starting distance to the optimal solution. If I understand correctly, your results involve a number of other unknown parameters, and hence I can only expect the shape of this hyperparameter optimization problem to be substantially more complex (despite the fact that in both cases this optimization problem depends on a sine parameter, the shape of this function can become more complex due to the presence of other hidden parameters). Perhaps numerical experiments can support the benefits of your analysis, but currently I don't see the adaptive nature of the results.
>
> **A1:** First, we want to emphasize again that this work focuses on theoretical analysis rather than proposing new algorithms. Thus the algorithms are still the original Adagrad, RMSprop and AMSgrad. Our “analysis” will not make these algorithms more complicated in terms of hyperparameter optimization. Second, the adaptive nature is reflected in the use of the dynamic covariance matrix $\hat V_t^{-1/2}$ rather than the learning rate $\alpha$. We call $\alpha \hat V_t^{-1/2}$ the effective learning rate since it is analogous to the classical learning rate in non-adaptive algorithms. Note that although $\alpha$ is a constant, the effective learning rate $\alpha \hat V_t^{-1/2}$ is still adaptive to the learning data, which makes our analyzed algorithms adaptive. Third, although the optimal choice of $\alpha$ depends on some problem-dependent parameters such as $T$ and $s$, in the worst case when $s=1/2$, the dependence of optimal $\alpha$ on $s$ can be removed and our analysis still gives an $O(\sqrt{d/T})$ convergence rate, which is better than the previous result $O(d/\sqrt{T})$ (Defossez et al., 2020). In practice, $\alpha$ is set as a constant and we can tune $\alpha$ to achieve a better convergence. This is exactly what practitioners are doing when using adaptive gradient methods. Therefore, the adaptivity nature of our analyzed algorithms will not be affected by our choice of $\alpha$.
>
> **Q2:**  Is it possible that you elaborate further on this in the paper?
>
> **A2:** We elaborate here why we can achieve a tighter dimension dependency ($d/\sqrt{T}$ v.s. $\sqrt{d}/\sqrt{T}$) as compared with Defossez et al., 2020. Both our analysis and the one in Defossez et al., 2020 required to upper bound the gradient norm $\|\nabla f(x_{\text{out}})\|\_2^2$ by the stochastic gradients $g_t$ and momentum $m_t$ (see our (6.9) and (A.19) in Defossez et al., 2020). However, Defossez et al., 2020 bounded $m_t$ and $g_t$ separately as illustrated in (A.20) in Defossez et al., 2020), and they obtained a better bound for $m_t$, which depends on $\alpha^2$, and a worse bound for $g_t$, which has an $\alpha^0 = O(1)$ dependency on $\alpha$. Thus, the final bound in their result suffers from an $\alpha^2d + d = O(d)$ dependency (see the second and third term in (A.54) in Defossez et al., 2020). In contrast, we bound both $m_t$ and $g_t$ by $\sum_{i=1}^d \|g_{1:T, i}\|_2$ uniformly by using Lemma 6.5, which makes our final bound only have an $\alpha^1 d$ term (see the third term in (6.10)). Therefore, by optimizing $\alpha$, our final bound only depends on $\sqrt{d}$ rather than $d$. We have added the explanation as Remark 6.6 and highlighted it in red color in our revision.
>
> **Q3:** Please add some explanatory details. By a quick look at the file I couldn't find these changes.
>
> **A3:** We have a new Lemma A.6 on page 15, which is the summary of several proof steps about how to bound $f(z_{t+1}) - f(z_t)$. We have now marked all the revisions in red for your convenient reference.
>
>
> **Q8:** Then I don't see how your rate matches the one in Ghadimi-Lan. That paper does not have any dimension dependence, whereas yours does.
>
> **A8**: Sorry for the confusion. We want to emphasize that our convergence rate matches the one by Ghadimi-Lan, 2013 in terms of the dependence on $T$, which is the number of iterations. For the dependency of dimension $d$, it is not directly comparable since they made a different stochastic noise assumption (they assumed the stochastic gradient is $\sigma$-subGaussian w.r.t. the 2-norm of the gradient). By directly translating their assumption to ours (replace $\sigma$ with $\sqrt{d}G_\infty$), we can obtain a $\sqrt{d/T}$ dominant term in their convergence result. We have also added this explanation in Remark 4.4 in the revision and highlighted them in red.

---

> ### Comment · Reviewer_6XWB · 2024-01-28
> **Final comments**
>
> Thank you for the clarifying comments. At a high level I have no further comments, but I just wanted to point out that you haven't resolved all the typos that I mentioned in my review. E.g. in page 18, there is still a line break in the middle of the probability that exp(\lambda Z_t) \geq exp(\lambda t).
>
> Having said that, I would still recommend acceptance (provided that the typos are corrected).

---

> ### Author Response · Authors · 2024-01-29
> **To Reviewer 6XWB's final comments**
>
> Thank you for your positive feedback and support! We have updated our paper by addressing all the typos you have mentioned.

---

### Review · Reviewer_Zzdv · 2024-01-11

**Summary Of Contributions:**

The authors provide convergence rates for three popular stochastic adaptive algorithms used in deep learning: AMSGrad, RMSProp, and AdaGrad.
Convergence rates are proved in expectation and with high probability for smooth non-convex functions under bounded stochastic gradients and sufficient control of the cumulative stochastic gradients assumptions.

**Audience:**

Yes

**Claims And Evidence:**

Yes

**Requested Changes:**

None

**Strengths And Weaknesses:**

Strengths:
The paper tackles a significant problem for the deep learning community.
The paper does a great job of introducing and explaining all the algorithms and the differences between each other.
In addition, the paper proposes a clear overview of the previous results.

Weaknesses:
I am not a leading expert in this area so I do not know how the paper is up to date with the recent literature on adaptive algorithms, this is my only "concern"

---

> ### Author Response · Authors · 2024-01-23
> **To Reviewer Zzdv**
>
> Thanks for your helpful feedback.
>
> **Q1:** I am not a leading expert in this area so I do not know how the paper is up to date with the recent literature on adaptive algorithms, this is my only "concern"
>
> **A1:** Thanks for your comment. Our result gives the state-of-the-art in-expectation and high probability convergence guarantees for adaptive algorithms. In our paper we have also compared our results with existing results.

---

### Author Response · Authors · 2024-01-23
**Paper revision**

Thanks for your constructive feedback. We have revised our paper according to your suggestions. Below are the changes in the revision.

1. We reorganized the proof structure. We summarize a Lemma A.6, which will be used by both the proof of Theorem 4.3 and 5.2, suggested by Reviewer 6XWB.
2. We added the experiment description about how to estimate the growth rate of the cumulative gradient $s$ to Appendix B.7, according to Reviewer gkXY.
3. We fixed several typos pointed out by Reviewer 6XWB.

---

### Decision · Action_Editor_WHBi · 2024-02-10

**Recommendation:** Accept as is

**Comment:**

Deep learning optimizers are the cornerstone of practical ML, and hence new works in this area should be of interest to many.

**Audience:**

The work is of a general interest to all interested in adaptive learning rates for deep learning optimizers AMSGrad, RMSProp and AdaGrad.

**Claims And Evidence:**

All reviewers agree that the presented results are sound, supported by evidence.